# Feedback to Reasoning: LLM-Assisted Molecular Optimization with Domain Feedback and Historical Reasoning

## Abstract

The remarkable success of large language models (LLMs) across diverse fields demonstrates their transformative potential in science, with molecular optimization representing a promising frontier. Traditionally, molecular optimization involves iterative discussions with domain experts, who progressively refine molecules with feedback until the desired properties are achieved. This interactive and feedback-driven process aligns well with the inherent strengths of LLMs, positioning them as promising tools for this task. **As an experience-driven task, molecular optimization depends critically on the domain feedback and accumulation of historical knowledge. However, none of the existing methods fully leverages such feedback and historical knowledge; especially, the reasoning trace and chemical insights that have led to successful optimization.** In this work, we propose **F2R**: Feedback to Reasoning, a conversational molecular optimization pipeline that allows LLMs to dynamically accumulate and retrieve historical knowledge about prior actions, rationales, and feedback. Moreover, just like humans whose reasoning is not always correct or precise, LLMs can also produce imperfect reasoning traces; F2R is the first work to leverages detailed domain feedback to critically reflect on and improve this reasoning. In this way, LLMs can evolve from passive language processors to agentic experts that emulate human experts in learning both actions and reasoning from experience. F2R is also the first work that leverages historical optimization results and reasoning traces from historical feedback. Consequently, F2R shows remarkable performance.

## 1 Introduction

The impressive capabilities of large language models (LLMs) in tackling a wide range of tasks have recently generated significant interest in extending their use to scientific fields, such as molecular optimization (Zheng et al., 2024; Zhang et al., 2025). Molecular optimization is inherently a complex, iterative process that relies heavily on expert input and continuous refinement. Given a molecule[1] and a target property, this process starts with consultations with domain experts who suggest possible molecular modifications using their domain experiences. The proposed changes are then implemented and the new analogue is tested in vitro (e.g., enzyme assays) or in silico (computational predictions) to see if the modification improves the desired properties. If the optimized molecule satisfies the target criteria, the process concludes. If not, expert chemists are informed by the test results and propose further refinement (Jorgensen, 2009; Cao et al., 2023; Liu et al., 2024). This cycle of iterative improvement fits naturally with one of the key strengths of LLMs, their capacity for interactive dialogue and incorporating feedback. Therefore, LLM-assisted conversational molecular optimization proceeds in a similar manner, with modifications given by LLMs. However, despite LLMs' remarkable generative and reasoning capabilities, they often still perform suboptimally on molecular optimization tasks. This suboptimality stems from the fact that LLMs lack the ability to accumulate and reuse historical knowledge and feedback from past tasks.

To leverage the broad underlying chemistry knowledge of LLMs, several studies have proposed integrating LLMs with specialized external guidance modules (agents) that provide optimization suggestions. These agent-assisted strategies aim to bridge the gap between the LLM's general chemistry

---

[1]Molecule refers broadly to both small molecules and larger macromolecules such as peptides and proteins.

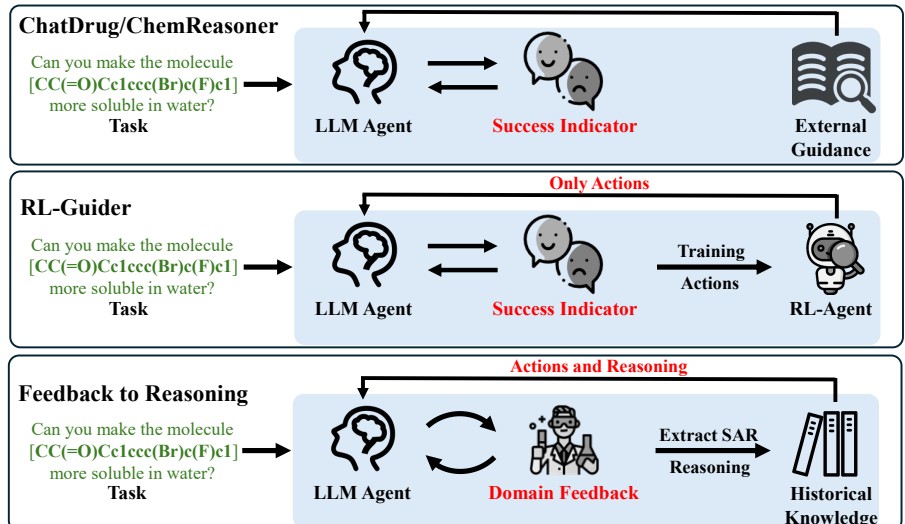

Figure 1: An overview of guidance pipelines for LLM-assisted molecule optimization. Importantly, prior works do not integrate domain feedback as guidance; instead, they only verify whether the optimized molecule meets the task objective. Furthermore, prior works either fail to incorporate historical experience (ChatDrug and ChemReasoner) or use it in a limited way that only provides guidance on the action to take without the reasoning behind it (RL-Guider). In contrast, Feedback to Reasoning (F2R) explicitly incorporates detailed domain feedback as guidance, enabling the LLM agent to self-reflect on its mistakes and refine its decisions. In addition, F2R systematically summarizes SAR patterns (see Sec. 4.2), accumulates historical knowledge during optimization, and retrieves the most relevant past experiences, providing guidance not only on the actions to take but also on the underlying reasoning.

knowledge and the specialized, context-dependent reasoning required for practical molecular optimization. Within a molecular optimization pipeline, each proposed candidate molecule is evaluated to determine its desirability. Human experts accumulate historical knowledge from this evaluation feedback, which includes information such as molecular validity and property values. Importantly, they reflect, reason, and learn why certain structural modifications succeed or fail, thereby building domain expertise that informs subsequent optimization efforts; this process of reflection and reasoning is crucial for domain experts. Nevertheless, existing guidance agent modules either fail to leverage any historical knowledge accumulation (Liu et al., 2024; Sprueill et al., 2024) or provide only static, single-action suggestions from a limited action space without considering the reasoning traces and implicit chemical knowledge (Liu et al., 2025b).

In this work, we propose F2R, a conversational molecular optimization pipeline that allows LLMs to dynamically accumulate and retrieve the rich historical knowledge about prior actions, rationales, and feedback while fully leveraging LLMs' broad general chemical knowledge and reasoning capabilities. An overview of F2R is provided in Fig. 1. **Overall, we summarize our contribution as follows:** ① We highlight the importance of enabling LLMs to learn from feedback, why past actions succeed or fail in each individual tasks, to better guide future actions (guided self-reflection). ② We propose F2R, a molecular optimization pipeline that enables guided self-reflection and dynamical historical reasoning for LLMs. ③ F2R does not rely on a predefined external knowledge base or a predefined action space. ④ Experiments demonstrate the superiority of F2R not only in quantitative evaluation metrics but also in transparent knowledge accumulation. ⑤We are the first to demonstrate very high success rates, sometimes exceeding 95%, on many tasks in the previously established test set of Liu et al. (2024). Thus, future work can reasonably omit these low-difficulty cases.

## 2 RELATED WORK

Over the past few years, machine learning has achieved remarkable success in learning from molecular data (Schütt et al., 2017; Chen et al., 2019; Jin et al., 2019; Duval et al., 2024). Moreover, in recent years, LLMs (Team et al., 2023; Achiam et al., 2023; Bi et al., 2024; Grattafiori et al., 2024) have demonstrated superior performance in various tasks, including, but not limited to, mathematical reasoning, code generation, and machine translation. The strengths of machine learning in molec-

ular learning and LLMs in various diverse tasks have sparked growing interest in leveraging LLMs for learning molecular tasks (Flam-Shepherd et al., 2022), including early works in fine-tuning language models for drug discovery (Cao et al., 2023; Liang et al., 2023). Molecular optimization (lead optimization, drug editing[2]) is a specific and important sub-task among drug discovery tasks; however, LLM-assisted molecular optimization remains underexplored. There are only a few works in this line of research. ChatDrug (Liu et al., 2024) employs a domain-specific database to retrieve similar known molecules that satisfy the desired properties as guidance. ChemReasoner (Sprueill et al., 2024) utilizes trained models with predefined domain knowledge to provide optimization suggestions. RL-Guider (Liu et al., 2025b) trains a reinforcement learning agent on past optimization results to offer guidance on which substructures to modify.

**Relations with Prior Works.** As mentioned in Liu et al. (2025b), both ChatDrug and ChemReasoner rely on a predefined knowledge base. Thus, they suffer from an inherent bias that the optimized molecules are overly similar to those present in the knowledge base. To mitigate these issues, RL-Guider (Liu et al., 2025b) leverages historical successes and failures to train a reinforcement learning agent with an action space restricted to a predefined set of atom types and functional groups. However, such a fixed action space limits exploration and constrains the reasoning capabilities of LLMs. Moreover, RL-Guider is trained on and provides guidance purely as discrete actions (e.g., replacing a hydroxyl group with an amino group). We argue that simply learning on actions is insufficient; LLMs need to understand the reasons behind successes and failures to learn historical experience and fully use their broad chemistry knowledge and reasoning abilities. Therefore, we propose F2R, a feedback-and-reasoning-enhanced pipeline that fully leverages this rich information, self-reflects on failed attempts, and paves the way for experience-driven, reasoning-aware LLM-assisted molecular optimization tasks. A comparison between F2R and existing work is provided in Table 1.

Table 1: A comparison of existing conversational molecular optimization pipelines.

| Pipeline | Utilization of Historical Knowledge | Free of Predefined Knowledge | Unconstrained Exploration Space | Guided Self-Reflection |
|---|---|---|---|---|
| Plain LLM | ✗ | ✓ | ✓ | ✗ |
| ChatDrug | ✗ | ✗ | ✗♣ | ✗ |
| ChemReasoner | ✗ | ✗ | ✓ | ✗ |
| RL-Guider | ✗◇ | ✗◇ | ✗ | ✗ |
| F2R (ours) | ✓ | ✓ | ✓ | ✓ |

♣ ChatDrug uses a limited external dataset as guidance; however, if the dataset is large enough, the space can be considered unconstrained.

◇ RL-Guider does not leverage any reasoning accumulated from past experience. It also uses a set of predefined actions, e.g., replacement of certain substructures.

# 3 BACKGROUND

## 3.1 MOLECULAR OPTIMIZATION WITH LLMS

Molecular optimization describes the task of transforming a given molecule into another that retains structural similarity while achieving a desired property. Formally, given an input molecule $x_{\text{in}}$ represented as a string and a textual task $x_{\text{t}}$ describing the optimization objective, molecular optimization can be formulated as a conditional generation problem (Liu et al., 2024) with the goal of producing an optimized molecule $x_{\text{out}} \sim P\left(x \mid x_{\text{in}}, x_{\text{t}}\right)$. In the context of LLM-assisted molecular optimization, this sampling from distribution $P$ is realized by a large language model that generates $x_{\text{out}} = \text{LLM}\left(x_{\text{in}}, x_{\text{t}}\right)$. In contrast to traditional deep generative models that are explicitly trained on domain-specific datasets, LLMs primarily rely on their general chemistry knowledge to reason about molecular structures and properties, enabling them to infer appropriate edits. As a result, LLMs are highly sensitive to the quality of the guidance they receive in the form of prompts, which shape their ability to reason effectively. In addition to their broad knowledge and reasoning capabilities, the conversational abilities (Bubeck et al., 2023) of LLMs are particularly valuable, making it natural for iterative tasks such as molecular optimization. When an edit is not successful, the user can provide feedback to the LLM and prompt it to refine its result. Formally, this is described as

$$x_i = \text{LLM}\left(x_{\text{in}}, x_{\text{t}}, x_{\text{g}}^i\right) \quad \text{for } i = 0, 1, 2, \ldots, K, \tag{1}$$

---

[2]In prior work, the term drug editing has been used. However, drug editing is a specific type of molecular optimization that focuses on therapeutics. Since this work also involves structural optimization tasks (such as peptide and protein optimization in Sec. 5), we use the broader term molecular optimization.

where $x_g^i$ denotes the guidance provided to the LLM at iteration $i$, $K$ is the maximum number of iterations, and the final output is $x_i$ for the smallest $i$ such that $x_i$ satisfies the target condition or once the maximum number of iterations is reached. The guidance can be as simple as informing the LLM that the generated molecule does not meet the requirements, or as complex as providing detailed instructions on how to adjust its reasoning. Existing work in this line of research aims to develop more effective forms of guidance for the LLM. A table of notations used in this work is provided in Appendix A.

## 3.2 MEMORY SYSTEMS FOR LLMs

Despite the remarkable success of LLMs across various fields, plain LLMs lack the ability to retain and leverage knowledge from previous interactions or external sources. This limitation constrains their capacity for tasks that require long-term context tracking, cumulative reasoning, or iterative improvement over time. To address this, memory systems (Packer et al., 2024; Wang et al., 2023; Edge et al., 2025) have been developed to extend the capabilities of LLMs by providing mechanisms for persistent storage and retrieval of relevant information to be added to the system prompt. These memory systems can be as simple as storing all past conversations across different sessions in a vector database indexed by their semantic embeddings or as complex as structured, self-organizing memory graphs or hierarchical memory modules (Chhikara et al., 2025; Xu et al., 2025). Nevertheless, existing memory mechanisms are primarily designed for general natural language tasks. Therefore, for molecular optimization tasks, designing an effective and specialized memory system for storing, updating, and retrieving historical knowledge accumulated through past optimization processes is crucial.

## 4 F2R: LEVERAGING HISTORICAL FEEDBACK AND REASONING

In this section, we introduce F2R, a novel conversational molecular optimization framework with historical knowledge and reasoning enhanced by feedback. While there exist general frameworks for memory and feedback mechanisms, such as OctoTools (Lu et al., 2025), F2R is designed specifically for LLM-assisted molecular optimization. It incorporates **domain-specific** similarity metrics for retrieval, summarizes **domain-specific** SAR patterns, and leverages **domain tools** to provide meaningful molecular feedback.

### 4.1 MOLECULAR OPTIMIZATION WITH FEEDBACK AND REASONING

Given an input molecule $x_{in}$ and a textual prompt $x_t$ describing the target, F2R proceeds in rounds:

$$x_0, a_0, r_0 = \text{LLM}\left(x_{in}, x_t, x_g^0 \| P_{edit}\right),$$
$$\gamma_{i-1}, x_i, a_i, r_i = \text{LLM}\left(x_{in}, x_t, x_g^i, f_{i-1} \| P_{edit}\right), \quad i = 1, 2, \dots, K, \tag{2}$$

where $f_i$ is the feedback on the optimized molecule from the $i$-th iteration ($x_i$); $P_{edit}$ is a carefully designed prompt template that requires the LLM to return not only the optimized molecule, but also the action taken ($a_i$) and the reasoning behind this particular edit ($r_i$); and $\gamma_{i-1}$ is the self-reflection on the last failed result $x_{i-1}$ based on the feedback $f_{i-1}$. This feedback can also be seen as a form of guidance that encourages the LLM to reason more critically. Here, equation 2 differs from equation 1 in two key aspects: ① the action and reasoning are explicitly required and structured in the output; and ② domain feedback (e.g., the optimize molecule failing to improve the desired property) is incorporated into subsequent iterations, enabling the model to self-reflect on its reasoning and refine future edits.

**Autonomous Domain Feedback.** While domain feedback can come from various sources, such as in vitro experiments (e.g., enzyme assays) or in silico predictions (computational models), our work focuses on a fully autonomous pipeline. This pipeline integrates computational software tools and rule-based algorithms to operate without human intervention. This process is described in Algorithm 1. Specifically, for optimized molecules, we first evaluate validity: if a molecule is invalid, the LLM receives detailed feedback on why it is invalid (`ExplainInvalidity`), which is done by combining computational tools and carefully designed prompt templates; if valid, its chemical properties are further analyzed and compared with those of the input molecule with respect to the optimization objective $x_t$. These evaluation feedback from computational tools are translated into natural language using carefully designed templates to ensure interpretability for LLMs (`ParseToNaturalLanguage`). Once the optimized molecule meets the objective, the

---

**Algorithm 1** Autonomous Domain Feedback

---

**Require:** Relevant property values of the input molecule $p_{\text{in}}$, optimization objective $x_{\text{t}}$, current optimized molecule $x_i$

1: **if** `IsValid`$(x_i)$ = **false then**
2:    $f_i \leftarrow$ `ExplainInvalidity`$(x_i)$
3:    **return** $f_i$, ContinueIteration
4: **end if**
5: $p_i \leftarrow$ `ComputeProperties`$(x_i)$
6: EvaluationFeedback$\leftarrow$ `ParseToNaturalLanguage`$(p_i, p_{\text{in}}, x_{\text{t}})$
7: **if** `MeetsObjective`$(p_i, x_t)$ **then**
8:    $f_i \leftarrow$ "The optimized molecule is valid." + EvaluationFeedback
9:    **return** $f_i$, StopIteration
10: **else**
11:    $f_i \leftarrow$ "The optimized molecule is valid." + EvaluationFeedback
12:    **return** $f_i$, ContinueIteration
13: **end if**=0

Note: ContinueIteration and StopIteration are Boolean indicators to continue and stop the optimization iterations, respectively.

---

iterative process terminates. For property values, we use computational software tools, including `RDKit` (Landrum et al., 2013) for small molecules, MHCflurry2.0 (O'Donnell et al., 2020) for peptides, and deep learning models such as ProteinDT (Liu et al., 2025a). For validity, we use `RDKit` for small molecules and rule-based algorithms for others; more details are described in Appendix B.1. An example of feedback obtained from interacting with `RDKit` is provided below:

> ***Example Feedback***
> *The optimized molecule: "CC[C@@](C)(NCC(=O)N(C)OC)c1nc(C)cs1O" is not valid.* Specifically, atoms at positions 13 (c), 14 (n), 15 (c), 17 (c), 18 (s), are aromatic, but there is no alternating single/double bonds that can be assigned to satisfy valence/electron count rules for those specific atoms.

### 4.2 AGENTIC AWARENESS OF STRUCTURE–ACTIVITY RELATIONSHIP

Structure–Activity Relationships (SARs) describe how the chemical structure of a molecule relates to its biological or pharmacological activity (e.g., solubility in water). Chemists design and synthesize related compounds, test their activities, and analyze how specific structural modifications affect performance. Overtime, SARs that chemists learn from past experience help them make informed decisions about which edits are likely to improve a molecule's properties. After an optimization task $j$ is finished, we further request the LLM to extract key SAR insights:

$$p, c = \text{LLM}\left(m || P_{\text{SAR}}\right), \tag{3}$$

where $m$ is the messages (iterations between the user and the LLM) in all the iterative rounds, $p$ denotes the extracted structural pattern or transformation (e.g., replace hydroxyl with amino), $c$ describes the conditions under which this pattern holds (e.g., only valid for aromatic rings), and $P_{\text{SAR}}$ is a carefully designed prompt template for the extraction of SAR patterns. All the prompts for F2R are provided in Appendix B.3.

### 4.3 LEVERAGING HISTORICAL EXPERIENCE

**Historical Knowledge Accumulation Module.** To accumulate historical experience, we store optimization histories in a structured way. Specifically, for an individual optimization task, we store a knowledge entry $e_j = \{x_{\text{in}}, x_{\text{t}}, p, c, (x_i, a_i, r_i, f_i, \gamma_i)_{\forall i}\}$. Overall, we will maintain a collection of knowledge entries $\mathcal{E} = \{e_1, e_2, \ldots, e_N\}$ as the historical knowledge base. For each optimization task performed by the LLM, a new entry is autonomously added to this growing knowledge base. As more edits are conducted across different molecules and targets, the accumulated knowledge collection $\mathcal{E}$ continues to expand over time.

**Historical Knowledge Retrieval Module.** As the historical knowledge base $\mathcal{E}$ expands over time, an important aspect of F2R is to effectively retrieve the most relevant pieces of accumulated knowledge to guide new optimization tasks. Given a new optimization task specified by $(x_{\text{in}}, x_{\text{t}})$, we iden-

tify a subset of historical knowledge entries that share the same target $\mathcal{E}_{x_t} = \{e_j \in \mathcal{E} \mid e_j[x_t] = x_t\}$, where $[\,\cdot\,]$ denotes accessing a specific field in the knowledge entry. Within $\mathcal{E}_{x_t}$, we further retrieve a set $R^*$ of $k_{\text{retrieval}}$ knowledge entries that are most relevant to the optimization task. Formally,

$$R^* = \underset{R \subseteq \mathcal{E}_{x_t}, |R| = k_{\text{retrieval}}}{\arg\max} \sum_{e \in R} \text{sim}\left(x_{\text{in}}, e[x_{\text{in}}]\right), \tag{4}$$

where $\text{sim}(\cdot, \cdot)$ is a similarity function that measures how similar the two molecules are. In practice, string representations of molecules (e.g., SMILES strings, protein sequences) are in discrete metric spaces. Standard similarity functions, such as cosine similarity, are no longer applicable. Tanimoto similarity is adopted for small molecules, and Levenshtein distance is adopted for peptides and proteins. More details about these similarity metrics are provided in Appendix B.2. Note that we only provide this guidance in the first round; in subsequent rounds, only feedback will be given. The SAR patterns in the retrieval knowledge entries $R^*$ will be used in the guidance $x_g^i$ in equation 2.

### 4.4 Why F2R? Feedback-driven Reasoning and Knowledge Accumulation

F2R offers unique advantages that make it particularly effective for molecular optimization tasks. We elaborate on two core advantages: *Feedback-driven Reasoning* and *Knowledge Accumulation*.

**Feedback-driven Reasoning.** In the molecule optimization process, each optimized molecule is assessed to determine whether it meets the desired properties, at which point the iterative process may conclude. When a molecule fails to satisfy these properties, the evaluation still yields valuable feedback, covering aspects such as validity, structural similarity, and chemical properties, that can assist human experts in refining their decisions and proposing more effective modifications. However, none of the existing methods[3] (Liu et al., 2024; 2025b) incorporates this feedback into the LLM-assisted iterative process. In contrast, F2R explicitly recognizes the importance of feedback and leverages it as a central component of reasoning. As a result, F2R learns from the feedback, corrects prior mistakes, and avoids repeated errors in subsequent iterations. As demonstrated by the experimental results in Sec. 5, guidance approaches that target reasoning (ChemReasoner and F2R) generally outperform those that focus on actions (RL-Guider and ChatDrug). Moreover, F2R surpasses ChemReasoner due to its feedback-driven reasoning mechanism.

**Knowledge Accumulation.** Chemists often build intuition by recognizing recurring structural patterns: similar molecules often exhibit similar behaviors. To capture this, F2R stores optimization trajectories and extracts transferable SAR patterns. Specifically, these patterns are not used merely to suggest a single action (Liu et al., 2025b) or retrieve a similar molecule (Liu et al., 2024); instead, they provide higher-level guidance that inspires the LLM to think about why a transformation may work and how it might be adapted to the new context. In this way, accumulated SAR knowledge serves as a source of reasoning, not just imitation. Importantly, LLMs might produce misleading reasoning that leads to incorrect results; these historical experiences and knowledge are self-reflected against the domain feedback, offering a more reliable source of knowledge and reasoning.

## 5 Experiments

In this section, we demonstrate the effectiveness of F2R through a series of experiments on various types of molecules, including small molecules, peptides, and proteins following (Liu et al., 2024). More details on these tasks are provided in Appendix C.

**Setup.** We compare F2R against the raw LLM without auxiliary guidance (Base LLM), ChatDrug, ChemReasoner, RL-Guider, and F2R itself. Similar to RL-Guider, F2R requires historical optimization results to demonstrate the effectiveness of knowledge accumulation. We construct a greedy coreset of small molecules, peptides, and proteins from the remaining dataset (excluding the test set). The size of this subset matches that of the test set. We perform molecular optimization tasks on this coreset as the initial historical results. For peptides and proteins, we do not compare with RL-Guider since it is not implemented in the original work. The proposed F2R method and all baselines are LLM-agnostic; we select GPT-4.1 and Gemini-2.5-Flash as the backbone models for evaluation. We follow the experimental setup described in Liu et al. (2024). Specifically, the maximum number of iterations, $K$, is set to 2. In the first round ($k = 0$), no guidance is provided. If the results are unsatisfactory, a second round is conducted with suggestions given to all methods. Likewise, if the

---

[3] RL-Guider (Liu et al., 2025b) incorporates feedback only as a reward signal for training the RL agent; the LLM itself does not receive feedback on the optimized molecules.

Table 2: Results on 16 single-objective small molecule optimization tasks. The best and second-best results are highlighted in red and blue, respectively. F2R consistently achieves the highest success ratios in 15 out of 16 tasks with ChatGPT-4.1 and 14 out of 16 tasks with Gemini-2.5-Flash. These results demonstrates the effectiveness of feedback-driven reasoning and knowledge accumulation.

| Task | Δ | ChatGPT-4.1 | | | | | Gemini-2.5-Flash | | | | |
|---|---|---|---|---|---|---|---|---|---|---|---|
| | | Base LLM | Chat Drug | Chem Reasoner | RL-Guider | F2R | Base LLM | Chat Drug | Chem Reasoner | RL-Guider | F2R |
| More soluble in water | 0 | 81.00 | 83.50 | 83.50 | 85.50 | 99.00 | 85.00 | 81.00 | 84.00 | 82.50 | 99.00 |
| | 0.5 | 84.00 | 81.50 | 84.00 | 83.50 | 96.00 | 80.50 | 81.50 | 76.50 | 79.50 | 96.00 |
| Less soluble in water | 0 | 85.00 | 85.50 | 84.50 | 85.50 | 99.00 | 95.50 | 97.00 | 98.00 | 91.50 | 99.00 |
| | 0.5 | 72.00 | 56.00 | 76.50 | 63.50 | 81.50 | 87.50 | 87.00 | 88.50 | 87.00 | 95.50 |
| More like a drug | 0 | 46.00 | 61.50 | 73.50 | 47.50 | 69.00 | 79.00 | 77.50 | 79.50 | 73.50 | 83.50 |
| | 0.5 | 6.00 | 20.00 | 18.00 | 8.50 | 21.00 | 16.50 | 27.00 | 22.50 | 19.50 | 30.50 |
| Less like a drug | 0 | 68.50 | 61.50 | 72.50 | 65.00 | 89.00 | 70.50 | 68.50 | 85.50 | 69.50 | 78.50 |
| | 0.1 | 16.50 | 28.50 | 52.00 | 24.50 | 63.50 | 44.00 | 43.00 | 67.00 | 53.50 | 65.00 |
| Higher permeability | 0 | 31.50 | 53.50 | 81.50 | 47.50 | 94.50 | 92.50 | 91.00 | 91.00 | 93.00 | 97.00 |
| | 10 | 19.50 | 36.50 | 62.50 | 34.00 | 74.00 | 52.50 | 62.00 | 63.00 | 61.50 | 79.00 |
| Lower permeability | 0 | 87.00 | 85.50 | 88.00 | 86.50 | 99.00 | 86.00 | 86.50 | 83.50 | 84.50 | 99.00 |
| | 10 | 87.00 | 83.50 | 88.50 | 86.50 | 97.50 | 85.00 | 81.50 | 82.00 | 84.50 | 98.50 |
| More hydro-bond acceptors | 0 | 74.00 | 69.00 | 76.50 | 77.50 | 97.00 | 80.50 | 82.50 | 78.50 | 74.50 | 99.00 |
| | 1 | 19.00 | 23.00 | 34.00 | 20.50 | 42.50 | 44.00 | 44.00 | 57.00 | 44.50 | 68.50 |
| More hydro-bond donors | 0 | 80.00 | 78.00 | 85.50 | 81.00 | 97.50 | 74.50 | 70.50 | 75.00 | 70.50 | 98.00 |
| | 1 | 13.00 | 26.50 | 19.50 | 22.50 | 41.50 | 16.50 | 15.00 | 47.00 | 15.50 | 52.50 |

results remain unsatisfactory, a third round is carried out. We follow the prompts exactly from Liu et al. (2024; 2025b) for the baseline methods, except that we additionally enforce compliance with the required format by employing *Structured Outputs* with a JSON schema.

**Evaluation Metric.** The performance is evaluated using the success ratio, defined as the proportion of generated molecules that are both valid and meet the desired target property, relative to the total number of optimization tasks. Note that this differs from the hit ratio used in Liu et al. (2024), where the ratio is computed as the number of successful results over the number of valid results. The success ratio is a stricter measure of performance also adopted in Liu et al. (2025b). The success ratio lower than hit ratio, particularly for small-molecule optimization tasks, as it is more common for LLMs to produce invalid SMILES strings.

## 5.1 CONVERSATIONAL MOLECULAR OPTIMIZATION WITH SMALL MOLECULES

We evaluate the performance of F2R against baseline methods on small molecules. The test molecules are sampled from the ZINC dataset, and following Liu et al. (2024), the evaluation is conducted on a set of 200 molecules. The molecular properties considered in this study can be deterministically computed using RDKit (Landrum et al., 2013).

**Evaluation.** An edited molecule is deemed successful if its property value improves by at least Δ compared to the original molecule. For instance, if Δ = 0.1 and the goal is to increase solubility, then the solubility of the modified molecule must exceed that of the original by at least 0.1 unit to be counted as a success. We evaluate under two settings: (1) *Single-objective optimization*, where the goal is to improve a single property value, and (2) *Multi-objective optimization*, where the goal is to improve two property values simultaneously.

**Results and Discussion.** We present the results for single-objective optimization in Table 2 and for multi-objective optimization in Table 4. We provide the visualization of an optimization task and the relevant history retrieved from historical knowledge in Table 3. Clearly, leveraging historical knowledge provides substantial guidance in accomplishing this task. In addition, in a case study presented in Appendix D.4, we showcase that feedback-driven reasoning helps the LLM correct its own mistakes (self-guided reflection). Clearly, F2R consistently outperforms the baseline methods. It achieves the highest success ratio in 15 out of 16 tasks with ChatGPT-4.1 and in 14 out of 16 tasks with Gemini-2.5-Flash for single-objective optimization. Moreover, it attains the highest success ratio across all multi-objective optimization tasks with both LLMs. Overall, we observe the trend that Gemini-2.5-Flash slightly outperforms ChatGPT-4.1, while being a smaller model for fast inference. This can be attributed to its reasoning capabilities; Gemini-2.5-Flash is a reasoning model with internal chain of thought (CoT) reasoning. This emphasizes the importance of reasoning in molecular optimization tasks.

Table 3: Visualization of an optimization task to make the input molecule *more like a drug* and the relevant SAR patterns from historical knowledge. Clearly, leveraging historical knowledge provides substantial guidance in accomplishing this task. We only show part of the SAR and reasoning due to length limits.

| | Input Molecule | Output Molecule | Reasoning |
|---|---|---|---|
| Historical Knowledge | CCCN(CCN) Cc1csc2ccccc12 | CCCN(CCN) Cc1csc2ccc(F)cc12 | SAR: 1. Addition of a second fluorine enhances metabolic stability and drug-likeness; 2. Chlorine substitution does not enhance drug-like...... |
| Current Optimization Task | NCC(CN)=CN[C@@H] (c1ccc(F)cc1)c1cccs1 | NCC(CN)=CN[C@@H] (c1ccc(F)cc1)c1ccc(F)s1 | Reasoning: Based on the history and SAR feedback, introducing a fluorine to the aromatic system is a proven strategy to enhance drug-likeness...... |

Table 4: Results on 12 multi-objective small molecule optimization tasks. The best and second-best results are highlighted in red and blue, respectively. F2R achieves the highest success ratios for all tasks. It should be noted that certain tasks are sufficiently challenging that all methods yield very low success ratios; however, F2R still demonstrates improvements on such a limited search space.

| Task | Δ | ChatGPT-4.1 | | | | | Gemini-2.5-Flash | | | | |
|---|---|---|---|---|---|---|---|---|---|---|---|
| | | Base LLM | Chat Drug | Chem Reasoner | RL-Guider | F2R | Base LLM | Chat Drug | Chem Reasoner | RL-Guider | F2R |
| More soluble in water | 0, 0 | 79.50 | 74.50 | 82.50 | 83.50 | 96.00 | 76.00 | 78.00 | 78.50 | 77.50 | 93.00 |
| More hydro-bond acceptors | 0.5, 1 | 20.00 | 26.50 | 39.00 | 28.50 | 40.50 | 38.50 | 31.00 | 44.00 | 36.50 | 59.50 |
| Less soluble in water | 0, 0 | 17.00 | 16.50 | 23.00 | 17.50 | 50.00 | 63.50 | 61.00 | 61.50 | 57.50 | 77.50 |
| More hydro-bond acceptors | 0.5, 1 | 0.00 | 4.00 | 0.50 | 1.00 | 4.50 | 13.50 | 13.00 | 14.00 | 13.00 | 25.00 |
| More soluble in water | 0, 0 | 86.00 | 84.00 | 86.00 | 88.50 | 98.50 | 73.00 | 72.00 | 70.00 | 70.50 | 96.50 |
| More hydro-bond donors | 0.5, 1 | 57.50 | 42.50 | 48.00 | 38.50 | 63.00 | 33.50 | 30.50 | 60.00 | 41.50 | 71.50 |
| Less soluble in water | 0, 0 | 4.00 | 16.50 | 28.00 | 21.00 | 54.50 | 64.00 | 64.00 | 65.50 | 64.50 | 78.50 |
| More hydro-bond donors | 0.5, 1 | 0.00 | 1.00 | 1.50 | 1.50 | 3.00 | 8.50 | 10.00 | 9.50 | 10.50 | 16.00 |
| More soluble in water | 0, 0 | 8.00 | 16.50 | 11.00 | 12.50 | 25.50 | 5.50 | 4.50 | 3.50 | 5.00 | 8.50 |
| Higher permeability | 0.5, 10 | 0.00 | 3.00 | 3.00 | 2.50 | 4.50 | 0.50 | 1.00 | 0.50 | 0.50 | 1.00 |
| More soluble in water | 0, 0 | 86.00 | 85.50 | 85.50 | 84.50 | 98.50 | 83.00 | 80.00 | 81.00 | 83.50 | 98.00 |
| Lower permeability | 0.5, 10 | 86.50 | 80.00 | 81.00 | 84.50 | 95.50 | 82.00 | 79.50 | 79.00 | 83.50 | 95.00 |

The performance of ChatDrug and RL-Guider does not show any consistent improvement over even the base LLM. We suspect that recent advanced LLMs possess sufficient chemical knowledge and reasoning capabilities such that external guidance on actions does not enhance reasoning; instead, it may mislead the model and even constrain its reasoning ability as the action is given. In contrast, ChemReasoner achieves the second best success ratio in most of the tasks because it leverages LLMs to reason and plan to achieve the optimization objective, thereby enabling guidance on explicit reasoning. This emphasizes the importance of guiding LLMs to better reasoning rather than providing static recommendations on actions to take.

### 5.2 CONVERSATIONAL MOLECULAR OPTIMIZATION WITH IMMUNOGENIC BINDING PEPTIDES

We evaluate the performance of F2R against baseline methods on immunogenic binding peptides. The test examples are sampled from the experimental dataset of peptide-MHC binding affinities (O'Donnell et al., 2020). This dataset contains 149 human MHC Class I proteins (alleles) and 309 thousand peptides. We randomly pick 500 target-source pairs from 30 common MHC proteins (alleles) following Liu et al. (2024) exactly.

**Evaluation.** The actual bindings require wet-lab experiments, which are expensive and prohibited for large-scale evaluation. MHCflurry2.0 (O'Donnell et al., 2020) is used as a pseudo-oracle to predict the peptide-MHC binding affinity. The success of peptide optimization must meet two criteria: (1) the resulting peptide should exhibit a higher binding affinity to the target allele than the original

peptide; and (2) the binding affinity between the edited peptide and the target allele must exceed a specified threshold. Following Liu et al. (2024), we set this threshold to be one-half of the average binding affinity observed in experimental data for the target allele. There are both single objective and multi-objective tasks; single objective tasks only require the peptide to bind to one target allele type, whereas multi-objective tasks require the peptide to bind to two target allele types. The detailed tasks are provided in Appendix C.

**Results and Discussion.** Similar to the case of small molecules, Gemini-2.5-Flash generally outperforms ChatGPT-4.1, underscoring the importance of reasoning. Across the benchmarks, F2R consistently achieves the best performance in 7 out of 8 tasks for both ChatGPT-4.1 and Gemini-2.5-Flash, while being a second in the remaining task. Although not as strong as F2R, ChemReasoner also demonstrates competitive results. Overall, these findings highlight the critical role of providing explicit reasoning guidance, thereby reinforcing the effectiveness of our method. This is a structural optimization task for binding, and it is non-trivial to visualize whether a structural change improves binding. However, a case study illustrating the optimization process for a peptide-optimization task is provided in Appendix D.4 to clarify the procedure.

Table 5: Results on 8 peptide optimization tasks. The task descriptions that correspond to these task IDs are provided in Appendix C. The best and second-best results are highlighted in red and blue, respectively. F2R consistently achieves the highest success ratios in 7 out of 8 tasks. These results demonstrate the generalizability of F2R to peptide tasks.

| Task | ChatGPT-4.1 | | | | Gemini-2.5-Flash | | | |
|---|---|---|---|---|---|---|---|---|
| | Base LLM | Chat Drug | Chem Reasoner | F2R | Base LLM | Chat Drug | Chem Reasoner | F2R |
| 301 | 3.40 | 62.80 | 84.80 | 95.60 | 93.80 | 94.60 | 96.20 | 98.20 |
| 302 | 46.60 | 40.60 | 51.60 | 55.80 | 60.20 | 51.00 | 64.00 | 72.20 |
| 303 | 2.60 | 52.80 | 61.80 | 66.80 | 77.80 | 85.40 | 90.20 | 87.20 |
| 304 | 53.80 | 35.80 | 59.60 | 64.00 | 43.20 | 40.00 | 45.20 | 56.40 |
| 305 | 47.00 | 34.80 | 56.20 | 63.80 | 50.40 | 43.80 | 50.40 | 67.80 |
| 306 | 25.20 | 61.60 | 80.60 | 78.20 | 79.60 | 87.20 | 90.00 | 95.40 |
| 401 | 28.20 | 15.40 | 27.60 | 34.40 | 19.80 | 14.60 | 15.60 | 26.80 |
| 402 | 10.40 | 12.00 | 11.80 | 13.80 | 16.00 | 10.80 | 13.80 | 17.60 |

## 5.3 Conversational Molecular Optimization with Protein Secondary Structures

We evaluate the performance of F2R against baseline methods on protein secondary structures. TAPE (Rao et al., 2019) is a benchmark for protein sequence property prediction, including the secondary structure prediction task. We use the test set of TAPE as our testing examples following Liu et al. (2024) exactly.

Table 6: Results on 2 protein optimization tasks. The best and second-best results are highlighted in red and blue, respectively. F2R achieves the highest success ratios in both tasks. These results demonstrate the generalizability of F2R to protein tasks.

| Task | ChatGPT-4.1 | | | | Gemini-2.5-Flash | | | |
|---|---|---|---|---|---|---|---|---|
| | Base LLM | Chat Drug | Chem Reasoner | F2R | Base LLM | Chat Drug | Chem Reasoner | F2R |
| More helix structures | 72.58 | 74.42 | 78.34 | 86.18 | 69.59 | 72.81 | 76.50 | 82.72 |
| More strand structures | 53.23 | 47.24 | 64.98 | 74.19 | 58.29 | 55.76 | 61.06 | 66.82 |

**Evaluation.** We use a pretrained secondary structure prediction model, ProteinCLAP-EBM-NCE from ProteinDT (Liu et al., 2025a), to evaluate the edited proteins. An edit is considered successful if the output protein sequences have more secondary structures than the input sequences.

**Results and Discussion.** The results are presented in Table 6. F2R consistently outperforms all baseline methods, further highlighting its generalizability to proteins. Consistent with the results on small molecules and peptides, ChemReasoner achieves the second-best performance, underscoring the importance of explicit guidance for reasoning. In Table 7, we also provide visualization of two optimization tasks alongside the LLM's reasoning, where retrieved historical knowledge offers substantial guidance in completing the tasks.

**Discussion on the Performance.** Overall, we have observed that LLMs, especially when equipped with F2R, perform very well on these molecular optimization tasks. Importantly, we are the first to demonstrate very high success rates (sometimes exceeding 95%) on many tasks in the previously

Table 7: Visualization of two optimization tasks aimed at producing *more helix structures* and *more strand structures*, respectively. The input protein is shown in light grey, and the final optimized protein is shown in red. In these examples, the LLM leverages relevant historical knowledge and learns from prior patterns to guide the optimization toward the desired secondary-structure enrichment.

| Task | Input&Output Proteins | Reasoning |
|------|----------------------|-----------|
| More Helix Structures |  | Based on the feedback from history, successful edits specifically increase helix content by targeting loop and coil regions (rich in glycine, proline, or serine) ) and strategically introducing helix-promoting residues such as alanine (A), leucine (L),...... |
| More Strand Structures |  | Based on prior feedback and structure-activity relationships, I specifically replaced or inserted multiple strand-favoring residues (V, I, F, Y, W, T) at positions not already dominated by these amino acids, particularly in regions...... |

established test set of Liu et al. (2024). Consequently, future work can reasonably omit these low-difficulty cases, as our results show. We discuss this in more detail and provide a reasonable list of updated tasks in Appendix D.1.

### 5.4 ADDITIONAL RESULTS AND VISUALIZATION

We conduct **ablation studies** to demonstrate that: ① Both feedback-driven reasoning and knowledge accumulation are crucial components of LLM-assisted molecular optimization pipelines; and ② As the knowledge base grows, the guidance it provides becomes more effective, leading to higher success rates. The results are provided in Appendix D.2. We provide failure analysis for all methods in Appendix D.3. We provide several case studies in Appendix D.4 that illustrate how feedback-driven reasoning and historical knowledge enable the LLM to make correct decisions. We also provide additional general visualization for all three types of optimization tasks (small molecules, immunogenic binding peptides, and protein secondary structures) in Appendix D.5.

## 6 CONCLUSION, LIMITATION, AND FUTURE WORK

In this work, we introduce F2R, a novel framework that employs a multi-agent system to autonomously accumulate, distill, and reuse historical knowledge and reasoning traces for more efficient and effective LLM-assisted molecular optimization. Specifically, F2R is the first to explicitly learn from detailed optimized outcomes, capturing not only actions but also the rationales behind successes and failures. By combining a dynamic historical knowledge base with an agentic SAR memory, F2R progressively refines its optimization strategies through experience-driven counterfactual replay, addressing key limitations of existing predefined guidance approaches. Experimental results across small molecules, peptides, and proteins demonstrate F2R's superior performance and transparent reasoning compared to strong baselines, validating the benefits of experience accumulation and agentic memory systems in complex molecular optimization tasks.

**Limitation and Future Work.** As a fully autonomous system relying on LLMs, F2R's effectiveness is ultimately limited by the reasoning accuracy and domain knowledge of the underlying language models. It remains a limitation and an avenue for future work to enhance the capabilities of the backbone LLMs. In particular, the proposed feedback mechanism and historical entries may provide a natural basis for reinforcement learning fine-tuning (RLFT), enabling models to iteratively refine their reasoning and domain adaptation. In addition, it could be of great potential to develop multi-agent systems for molecular optimization that allow the interaction of LLMs and various domain tools in a collaborative ecosystem, collectively advancing the efficiency and reliability of autonomous discovery. In addition, it is interesting to examine cases where incorrect reasoning leads to successful results, or where correct reasoning does not, and whether these situations cause error accumulation and propagation.

ETHICS STATEMENT

This research focuses on the development and evaluation of LLM-assisted molecular optimization methods, leveraging LLMs to optimize molecules as part of the drug discovery process. The study does not involve human subjects, personal data, or sensitive information that could raise privacy, security, or fairness concerns. No potential conflicts of interest, legal compliance issues, or harmful applications have been identified in this work.

REPRODUCIBILITY

We contain necessary details to reproduce our results in the paper, including the prompts used. The datasets employed are entirely open-source as detailed in the appendix. All code, datasets, and instructions necessary for reproduction will be made publicly available upon acceptance.

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

## LLM Usage

LLMs are used only at the sentence level to help improve readability and reduce typos and grammatical errors. LLMs are also used as coding agents (Coding CoPilot) in the programming process. All LLM-generated objects, including code, text, suggestions, etc., have been carefully verified by the authors before use.

## A  Table of Notations

We summarize notations used throughout this paper in Table 8.

Table 8: Summary of Notations.

| Notation | Description |
|---|---|
| $x_{\text{in}}$ | Input molecule (SMILES strings, peptide and protein sequences). |
| $x_{\text{t}}$ | Textual task or optimization objective. |
| $x_{\text{out}}$ | Final optimized molecule produced by the LLM. |
| $K$ | Maximum number of allowed optimization iterations. |
| $x_i$ | Optimized molecule at iteration $i$. |
| $x_{\text{g}}^i$ | Guidance (feedback, SAR knowledge, etc.) provided to the LLM at iteration $i$. |
| $a_i$ | Action taken by the LLM at iteration $i$ (e.g., structural edit). |
| $r_i$ | Reasoning provided by the LLM at iteration $i$ (why the action was taken). |
| $f_i$ | Domain feedback on the optimized molecule $x_i$ (validity, property change, etc.). |
| $\gamma_i$ | Self-reflection of the LLM at iteration $i$, based on $f_i$. |
| $P_{\text{edit}}$ | Structured prompt template requiring action and reasoning output. |
| $p$ | Extracted structural pattern (e.g., hydroxyl $\rightarrow$ amino substitution). |
| $c$ | Condition under which a structural pattern $p$ holds (e.g., for aromatic rings). |
| $m$ | All iterative messages (dialogue history) between user and LLM in an optimization task. |
| $e_j$ | Knowledge entry for optimization task $j$: $\{x_{\text{in}}, x_{\text{t}}, p, c, (x_i, a_i, r_i, f_i, \gamma_i)_{\forall i}\}$. |
| $\mathcal{E}$ | Historical knowledge base of all accumulated entries. |
| $\mathcal{E}_{x_{\text{t}}}$ | Subset of $\mathcal{E}$ containing tasks with the same target $x_{\text{t}}$. |
| $k_{\text{retrieval}}$ | Number of historical entries retrieved. |
| $R^*$ | Retrieved subset of $k_{\text{retrieval}}$ most relevant entries for a new task. |
| $\text{sim}(\cdot, \cdot)$ | Similarity function (Tanimoto similarity for small molecules, Levenshtein distance for peptides/proteins). |
| $p_{\text{in}}$ | Property values of the input molecule. |
| $p_i$ | Property values of the optimized molecule $x_i$. |

## B  Additional Details for F2R

### B.1  Validity and Feedback

**Small Molecules.** For small molecules, we use the `Chem.MolFromSmiles()` method from `RDKit`. First of all, we detect if there are characters that are not possible to appear in a SMILES string by regular expression (rule-based). Then, if the SMILES string contains only valid characters, we use RDKit to test for validity. In particular, there are several common reasons for it to fail. ① Kekulization failures: There are aromatic atoms that can't be assigned alternating single/double bonds. ② Valence errors: Atom exceeds its allowed valence. ③ Caromaticity assignment: Mixing aromatic and non-aromatic atoms in a ring inconsistently. ④ Ring problems: Missing ring closure digits (unclosed rings). We carefully design templates to convert the error messages into natural

language for LLM feedback. It is rare, but there are other potential errors, we directly use the error message as the feedback in such case.

**Peptides and Proteins.** Peptides and proteins are represented as sequences of amino acids using the standard one-letter code. Among the 26 English alphabets, 20 correspond to the canonical amino acids, 2 (U and O) represent rare amino acids, and 4 (B, Z, J, X) are reserved for ambiguous or unknown residues. In principle, any combination of these letters forms a syntactically valid sequence, although not all combinations correspond to chemically plausible or biologically functional proteins. For validation, we detect invalid characters using regular expressions. If only valid letters are present, the sequence is accepted as a valid peptide/protein string. Depending on the task, we can restrict to only the canonical amino acids or unambiguous residues. Additional biological plausibility checks (e.g., domain detection or similarity search) may also be applied depending on the downstream task.

## B.2 Tanimoto Similarity and Levenshtein Distance

**Small Molecules.** For small molecules represented in SMILES, Tanimoto similarity measures the similarity between two molecules based on their chemical First, SMILES strings are converted into molecular fingerprints (binary) by RDKit. Then, the Tanimoto similarity coefficient is defined as:

$$S(A, B) = \frac{|A \cap B|}{|A \cup B|} \in [0, 1],$$

where $A$ and $B$ are two binary fingerprint vectors, $|A \cap B|$ is the number of common bits set to 1 in both fingerprints, $|A \cup B|$ is the number of bits to 1 in either fingerprint.

**Peptides and Proteins.** For peptide and protein sequences, the Levenshtein distance provides a way to measure sequence similarity by counting the minimum number of edit operations required to transform one amino acid sequence into another. The Levenshtein distance between two amino acid sequences $a$ and $b$ (of lengths $|a|$ and $|b|$ respectively) is defined as $\mathrm{lev}(a, b)$:

$$\mathrm{lev}(a, b) = \begin{cases} |a| & \text{if } |b| = 0 \\ |b| & \text{if } |a| = 0 \\ \mathrm{lev}(\mathrm{tail}(a), \mathrm{tail}(b)) & \text{if head } (a) = \mathrm{head}(b), \\ 1 + \min \begin{cases} \mathrm{lev}(\mathrm{tail}(a), b), \\ \mathrm{lev}(a, \mathrm{tail}(b)), \\ \mathrm{lev}(\mathrm{tail}(a), \mathrm{tail}(b)) \end{cases} & \text{otherwise.} \end{cases}$$

Here, $\mathrm{head}(x)$ denotes the first character of the sequence $x$ (i.e., the first amino acid), and $\mathrm{tail}(x)$ is the remainder of the sequence after removing the first character. The recurrence relation captures three types of edit operations: (1) Deletion: removing an amino acid from $a$ to align with $b$; (2) Insertion: adding an amino acid into $a$ to align with $b$; (3) Substitution: replacing one amino acid in $a$ with another to match $b$. Thus, the Levenshtein distance $\mathrm{lev}(a, b)$ corresponds to the minimum number of insertions, deletions, and substitutions needed to transform sequence $a$ into sequence $b$.

## B.3 Prompts

We give an example of conversation messages for F2R for a complete optimization round.

We have the following system prompt following prior work:

> **System Prompt**
>
> You are a helpful chemistry expert with extensive knowledge of drug design.

In the first round, we ask the LLM to generate 5 edited candidates following the exact setup as in Liu et al. (2024), and the first that does not contain invalid characters will be taken as the edited drug. Note that we now require the reasoning traces and edited drugs to be structured as a json object through *Structured Outputs* with a JSON schema. All baselines adopt this strategy to ensure that the edited drugs are structured.

> **First Round Task Prompt**
>
> <task_prompt>. The output <drug_type> should be similar to the input molecule. Give me 5 <drug_type> in string representation only.
> Give me a short reasoning first, and then a listed of <drug_type> under the key "edited_drugs".
> Return the result as a json object in the following format:
> {{"reasoning": <reasoning>,
> "edited_drugs": [<string1>, <string2>, <string3>, <string4>, <string5>] }}

The task prompts are given in details in Appendix C below. If the edited drug in the first round is not valid or does not satisfy the property requirements:

> **Iterative Round Task Prompt**
>
> The generated drug <edited_drug> is evaluated: <autonomous_domain_feedback>.
> Here is the history and feedback from the most similar molecule for the same task:
> - The original drug is: <input_drug>
> - The edited drug is: <edited_drug>
> - The feedback on the attemp from evaluation: <autonomous_domain_feedback>
> - Potential structure-activity relationship summarized from the same task: <SAR_patterns>
> However, you should remember that it does not always apply to your task.
> Can you give me a new <drug_type>? This time, return only one <drug_type> in string format.
> Return the result as a json object in the following format:
> {{ "reasoning": <reasoning>,
> "edited_drugs": [<string1>] }}

After the task in completed or the maximum number of iterations is reached, the LLM is asked to summarize the conversation and identify transferable SAR patterns:

> **SAR Task Prompt**
>
> 1. Summarize in concise sentences why these edits were successful or not. This should also include certain reasoning and how to improve. Be explicit and relatively short.
> 2. If you see any clear structure–activity relationship (SAR) trend across different molecules, state in precise and very short phrases. If you do not see any obvious SAR pattern, just return an empty string.
> Return ONLY the following JSON object:
> {{ "summary": <concise bullet or sentence per peptide, keep in a single string>,
> "sar": [<very short SAR statement or empty string>] }}

## C    TASK DESCRIPTION AND TASK PROMPTS

**Small Molecules.** For small molecules, the task is to edit the SMILES string to satisfy the task requirements listed in Table 2 and Table 4 respectively (e.g. "more soluble in water"). The task prompt is:

> **Task Prompt for Small Molecules**
>
> Can you make the molecule <input_smiles_string> <task_requirements>?

**Peptides.** For peptides, the task is to modify a peptide that binds to a source allele type so that it can bind to given target allele type(s), which is a common task in peptide design and immunology research. The task prompt is:

---

**Task Prompt for Peptides**

We want a peptide that binds to <target_allele_types>. We have a peptide <input_peptide> that binds to <source_allele_type>, can you help modify it?

---

The task IDs and their corresponding source and target allele types are given in Table 9.

Table 9: Target allele type(s) and source allele type for peptide optimization.

| Task ID | Source Allele Type | Target Allele Type(s) |
|---------|--------------------|-----------------------|
| 301 | HLA-C*16:01 | HLA-B*44:02 |
| 302 | HLA-B*08:01 | HLA-C*03:03 |
| 303 | HLA-C*12:02 | HLA-B*40:01 |
| 304 | HLA-A*11:01 | HLA-B*08:01 |
| 305 | HLA-A*24:02 | HLA-B*08:01 |
| 306 | HLA-C*12:02 | HLA-B*40:02 |
| 401 | HLA-A*29:02 | HLA-B*08:01 and HLA-C*15:02 |
| 402 | HLA-A*03:01 | HLA-B*40:02 and HLA-C*14:02 |

**Proteins.** For proteins, the task is to modify a protein sequence so that more amino acids adopt desired secondary structures, specifically $\alpha$-helix (more helix structure) or $\beta$-strand (more strand structure) conformations. The task prompt is:

---

**Task Prompt for Small Molecules**

We have a protein <input_protein_sequence> <. Can you modify it by making more amino acids into the <desired_secondary_structure_type> (secondary structure)?

---

# D  ADDITIONAL EXPERIMENTAL RESULTS AND DISCUSSION

## D.1  DISCUSSION ON SATURATED RESULTS

We follow the experimental setup as well as the exact test tasks introduced in the pioneering work ChatDrug (Liu et al., 2024). However, as we have shown in Sec. 5, LLMs equipped with reasoning and memory capabilities can achieve very high success rates on many of these tasks with the *recent development of powerful LLMs*. These saturated results suggest that the current set of test problems may need to be updated. At its current stage, LLM-assisted molecular optimization is not intended to replace human experts; rather, we aim to explore the potential and limits of LLMs in chemical reasoning and in performing these tasks. Therefore, it is important to develop a representative test set that is sufficiently challenging and covers a broader range of problem types. As LLMs continue to scale, gain power, and exhibit advanced reasoning capabilities, the overall cost, computationally or monetarily, required to use them also rises. The test set introduced in ChatDrug, which consists of more than $10,000$ individual optimization tasks, becomes unaffordable if performed on advanced models. Therefore, it is also important to limit the number of individual optimization tasks. To address this need, we advocate for a reduced task set based on our experimental results and observations. For all optimization tasks where F2R achieves a success rate above $90\%$, future work should randomly sample and retain only $10\%$ of those tasks. For tasks with a success rate above $80\%$, future work should keep a randomly sampled subset consisting of $20\%$ of the tasks. Future work should combine these subsets into three tasks based on molecular type (Small Molecule – Easy, Peptide – Easy, or Protein – Easy) and report the results jointly for each type.

## D.2 Ablation Study

**Separating Feedback-Driven Knowledge and Historical Knowledge.** We conduct an ablation study to disentangle the effects of feedback-driven reasoning and historical knowledge from knowledge accumulation. Specifically, we use only the feedback (F2R-Feedback) as guidance and only the historical knowledge entries (F2R-Hist.) as guidance, respectively. We select the last four single-objective small molecule optimization tasks where F2R significantly outperforms the base LLM, allowing us to clearly attribute the observed performance gains. The results are presented in Table 10. Clearly, both contribute to the superior performance of F2R; feedback accounts for more substantial performance gains, but historical knowledge also plays a critical role that further improves performance.

Table 10: Ablation study on 4 single-objective small molecule optimization tasks with Gemini-Flash-2.5. Clearly, both feedback driven reasoning and historical knowledge accumulation are important to the superior performance fo F2R.

| Task | $\Delta$ | Base LLM | F2R-Hist. | F2R-Feedback | F2R |
|---|---|---|---|---|---|
| More hydro-bond acceptors | 0 | 80.50 | 86.50 | 94.00 | 99.00 |
| | 1 | 44.00 | 50.50 | 58.50 | 68.50 |
| More hydro-bond donors | 0 | 74.50 | 78.50 | 91.50 | 98.00 |
| | 1 | 16.50 | 39.00 | 45.50 | 52.50 |

**Size and Retrieval Count of Historical Knowledge.** In this ablation study, we investigate how the size of the historical knowledge database and the number of retrieved entries influence the model's performance. Given the cost of extensive experimentation, we perform this ablation study on only the first task introduced above (More hydro-bond acceptors with $\Delta = 0$). We test with 3 different historical knowledge sizes 100, 200, and 500 (200 is used for all small molecule optimization tasks in the main paper), and 3 different retrieval counts 1, 3, and 5 (3 is used for all small molecule optimization tasks in the main paper). In addition, we do not provide domain feedback in this experiment; only historical knowledge is used to control the variable. The results are provided in Table 11. These results indicate that enlarging the historical knowledge dataset leads to clear improvements in success rate. However, increasing the retrieval count does not reliably enhance performance, especially when the available historical knowledge remains limited. This is likely because when the database is small, it contains relatively few relevant historical examples, and retrieving unrelated entries may introduce distracting or misleading information, ultimately limiting the benefit of larger retrieval counts.

Table 11: Ablation study on the impact of size of the historical knowledge database ($N$) and the number of retrieved entries ($k_{\text{retrieval}}$) for the small-molecule optimization task "More hydro-bond acceptors" with $\Delta = 0$, using Gemini-Flash-2.5.

| $k_{\text{retrieval}} \backslash N$ | 100 | 200 | 500 |
|---|---|---|---|
| 1 | 80.50 | 82.50 | 83.00 |
| 3 | 84.50 | 86.50 | 86.50 |
| 5 | 84.00 | 86.00 | 87.50 |

## D.3 Failure Analysis

We provide the reasons for failure for each method to offer another perspective on the sources of improvement achieved by F2R. For peptides and proteins, the primary source of failure comes from not meeting the task requirements (e.g., the optimized peptide does not bind to the target allele type, or the optimized protein does not contain more amino acids in the desired secondary structures). There are almost no cases in which an invalid peptide or protein is produced. Therefore, the main benefit of feedback and memory in F2R is to guide the LLM toward satisfying the task requirements. For small molecule optimization, we find that LLMs often make invalid SMILES strings, which can be seen in the failure count of each method in Table 12. Clearly, methods incorporating reasoning (F2R

and ChemReasoner) demonstrate strong performance in meeting property requirements. However, ChemReasoner does not reduce SMILES syntax errors. In contrast, F2R achieves both satisfactory property values and significantly fewer syntax errors, thanks to the use of feedback and memory. Note that ChemReasoner appears to have fewer failures due to not satisfying property requirements; however, this should be interpreted with caution, as it also produces a large number of syntax errors. These syntax errors may arise in difficult tasks where the model attempts extreme edits that ultimately violate SMILES grammar.

Table 12: Failure analysis of all methods for small-molecule optimization with $\Delta = 0$ using ChatGPT-4.1. The numbers are reported in the format S/P/I, where S denotes the number of successes, P denotes the number of failures due to not satisfying property requirements, and I denotes the number of failures caused by invalid SMILES strings. Clearly, methods incorporating reasoning (F2R and ChemReasoner) demonstrate strong performance in meeting property requirements. However, ChemReasoner does not reduce SMILES syntax errors. In contrast, F2R achieves both satisfactory property values and significantly fewer syntax errors, thanks to the use of feedback and memory.

| Task | F2R | BaseLLM | ChatDrug | ChemReasoner | RL-Guider |
|---|---|---|---|---|---|
| More soluble in water | 198 / 1 / 1 | 162 / 2 / 36 | 167 / 3 / 30 | 167 / 0 / 33 | 171 / 1 / 28 |
| Less soluble in water | 198 / 0 / 2 | 170 / 1 / 29 | 171 / 4 / 25 | 169 / 0 / 31 | 171 / 1 / 28 |
| More like a drug | 138 / 51 / 11 | 92 / 73 / 35 | 123 / 49 / 28 | 147 / 24 / 29 | 95 / 71 / 34 |
| Less like a drug | 178 / 16 / 6 | 137 / 22 / 41 | 123 / 32 / 45 | 145 / 14 / 41 | 130 / 24 / 46 |
| Higher permeability | 189 / 6 / 5 | 63 / 86 / 51 | 107 / 50 / 43 | 163 / 7 / 30 | 95 / 56 / 49 |
| Lower permeability | 198 / 0 / 2 | 174 / 0 / 26 | 171 / 5 / 24 | 176 / 0 / 24 | 173 / 8 / 19 |
| More hydro-bond acceptors | 194 / 2 / 4 | 148 / 2 / 50 | 138 / 7 / 55 | 153 / 0 / 47 | 155 / 4 / 41 |
| More hydro-bond donors | 195 / 2 / 3 | 160 / 2 / 38 | 156 / 8 / 36 | 171 / 0 / 29 | 162 / 14 / 24 |
| **Total** | **1488 / 78 / 34** | **1106 / 188 / 306** | **1156 / 158 / 286** | **1291 / 45 / 264** | **1152 / 179 / 269** |

## D.4 CASE STUDIES: FEEDBACK DRIVEN SELF-REFLECTION AND HISTORICAL KNOWLEDGE

We provide several case studies to showcase instances where feedback-driven reasoning and historical knowledge lead to successful optimization, whereas without them, the LLM struggles to produce a satisfying molecule. One case study is provided for each type of molecule (small molecules, peptides, and proteins), using either feedback-driven reasoning or historical knowledge to illustrate the optimization process. However, we note that for all molecule types, both feedback-driven reasoning and historical knowledge significantly aid the LLM in performing these tasks.

### D.4.1 FEEDBACK-DRIVEN REASONING

We first provide a case study that showcases instances where the LLM initially produces an invalid molecule, and we then provide detailed feedback to the LLM. We also perform the same experiment without detailed feedback from domain tools, using only a success indicator as done in prior works. To do this, we reuse the same messages until feedback so that the input molecule and the optimized molecule from the first iteration are the same. In this case, the LLM makes the same mistake of producing an invalid molecule.

With Feedback:

- **Input Molecule:** *Cn1ccc(C(=O)Nc2sc3c(c2C#N)CCC3)cc1=O*
- **Task:** *More soluble in water*
- **Optimized Molecule (1st iteration):** *HOCH2c1cc(Sc2cc3ccccc3[nH]2)ccc1C=O* (This is an invalid molecule due to a syntax error in the methylene group)
- **Feedback:**
  *The generated drug HOCH2c1cc(Sc2cc3ccccc3[nH]2)ccc1C=O is checked by experts and it is NOT valid because of RDKit error: "syntax error while parsing; check for mistakes around position 1; failed to parse SMILES."*
- **Reasoning:**
  *The previous SMILES HOCH2c1cc(...) was syntactically incorrect for a hydroxymethyl*

*group. The corrected SMILES replaces the methyl group with a hydroxymethyl group (CH2OH) by explicitly defining the bond from the ring carbon to the new carbon, which is then bonded to oxygen, thereby increasing water solubility through the addition of a hydroxyl group.*

- **Optimized Molecule (2nd iteration):** *c1(CO)cc(Sc2cc3ccccc3[nH]2)ccc1C=O* (This is a valid molecule that is more soluble in water)

Without Feedback:

- **Input Molecule:** *Cn1ccc(C(=O)Nc2sc3c(c2C#N)CCC3)cc1=O*
- **Task:** *More soluble in water*
- **Optimized Molecule (1st iteration):** *HOCH2c1cc(Sc2cc3ccccc3[nH]2)ccc1C=O* (This is an invalid molecule due to a syntax error in the methylene group)
- **Feedback:**
  *Your provided sequence HOCH2c1cc(Sc2cc3ccccc3[nH]2)ccc1C=O is not correct.*
- **Reasoning:**
  *A small polar modification near the aromatic core can improve water solubility while retaining overall scaffold similarity. Introducing a hydroxyl-bearing substituent increases hydrogen-bonding capacity without drastically changing the molecule's shape.*
- **Optimized Molecule (2nd iteration):** *OCH2Cc1cc(Sc2cc3ccccc3[nH]2)ccc1C=O* (This is still an invalid molecule due to a syntax error in the methylene group)

Clearly, in this example, the LLM is able to identify and correct its own mistake once explicit structural feedback is provided. With guidance, the model transitions from producing an invalid SMILES string to generating a syntactically valid molecule that reflects the intended functional modification. In contrast, without feedback, the LLM repeats the same mistake, the syntax error in the methylene group remains uncorrected. This demonstrates the effectiveness of feedback-driven reasoning in guiding the model toward correct outputs..

### D.4.2 HISTORICAL KNOWLEDGE

We present two cases showing the role of historical knowledge in optimization: an example where the model benefits from prior successes, and another example where it learns from past failures to avoid repeating the same errors.

Learning from Past Success:

- **Input Peptide:** *ITAPTVRQY*
- **Task:** *We want a peptide that binds to HLA-B\*44:02. We have a peptide ITAPTVRQY that binds to HLA-C\*16:01, can you help modify it?*
- **Optimized Peptide (1st iteration):** *IPAPTVRQY* (This is a valid peptide; however, it does not bind to HLA-B\*44:02)
- **Historical Knowledge:**
  *Original Peptide: AEAGVRLLSY; Optimized Peptide:AEAGVRLLSW; Summary: Substitution to W at the C-terminus enhanced binding to HLA-B\*44:02*
- **SAR:**
  *C-terminal tryptophan (W) substitution improves HLA-B\*44:02 binding*
- **Reasoning:**
  *The feedback from 'History 2' indicated that substituting the C-terminal Tyrosine (Y) with Tryptophan (W) significantly enhanced binding to HLA-B\*44:02 (AEAGVRLLSY → AEAGVRLLSW). Given that the current peptide ITAPTVRQY also has a Tyrosine at its C-terminus, replacing it with Tryptophan is a direct application of a previously successful modification to improve HLA-B\*44:02 binding.*
- **Optimized Peptide (2nd iteration):** *ITAPTVRQW* (This is a valid peptide, and it binds to HLA-B\*44:02)

Learning from Past Failure:

- **Input Protein:** *MRGSHHHH......* (Lengthy protein Sequence)
- **Task:** *Making more amino acids into the strand structure (secondary structure)*
- **Optimized Protein (1st iteration):** *DQVGKSPN......* (It does not have more amino acids in the strand structure)
- **Historical Knowledge:**
  *Edits 1, 2, and 5 failed to increase strand-forming amino acids likely due to minor side-chain substitutions or loop/helix-preserving changes......*
- **SAR:**
  *Fluorine substitution and heteroaromatic ring incorporation improve drug-likeness*
- **Reasoning:**
  *Based on prior feedback and histories, simply substituting residues one by one is not always sufficient to promote additional strand (beta-sheet) structure. To improve strand (beta-sheet) propensity based on the feedback, I applied more extensive and targeted substitutions in predicted loop/coil regions and at the edges of existing beta sheets, using amino acids with a strong tendency to promote beta-strand conformation (V, I, F, Y, T, Q, E, D, A, G). I emphasized patchwise replacement rather than single-residue changes, aligning stretches of V, I, F, Y for extended backbone and avoiding substitutions in core (likely functionally essential) regions.*
- **Optimized Protein (2nd iteration):** *DQVGKSPN......* (It has more amino acids in the strand structure)

## D.5 ADDITIONAL VISUALIZATION OF RESULTS

We provide additional visualization for all three types of optimization tasks—small molecules, immunogenic binding peptides, and protein secondary structures—to offer a clearer qualitative and visual understanding of the results of LLM-assisted molecular optimization. Specifically, the visualization of input and optimized small molecules are shown in Fig. 13. For peptides, we provide the visualization using position weight matrices (PWMs) in Fig. 2. PWMs have been widely used for the visualization of protein motifs (patterns), and they plot the distribution of each amino acid at the corresponding position. Thus, more important motifs with higher probabilities will be marked with higher letters. The visualization of input and optimized proteins are shown in Fig. 14.

Table 13: Visualization of six small molecule optimization tasks.

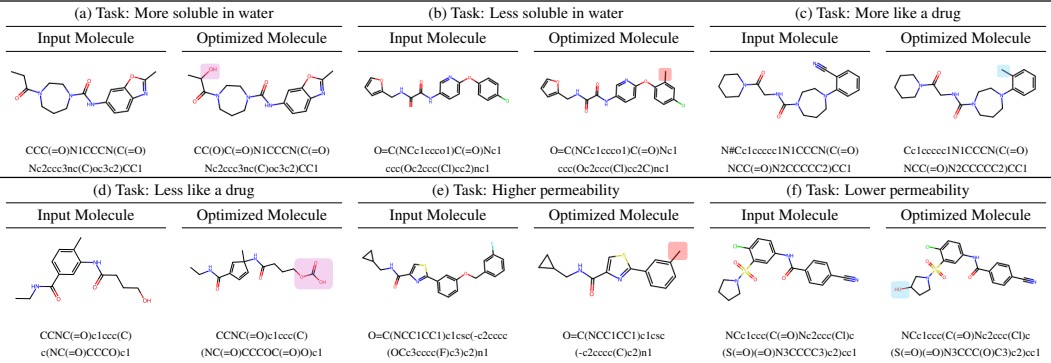

| (a) Task: More soluble in water | | (b) Task: Less soluble in water | | (c) Task: More like a drug | |
|---|---|---|---|---|---|
| Input Molecule | Optimized Molecule | Input Molecule | Optimized Molecule | Input Molecule | Optimized Molecule |
| CCC(=O)N1CCCN(C(=O)Nc2ccc3nc(C)oc3c2)CC1 | CC(O)C(=O)N1CCCN(C(=O)Nc2ccc3nc(C)oc3c2)CC1 | O=C(NCc1ccco1)C(=O)Nc1ccc(Oc2ccc(Cl)cc2)nc1 | O=C(NCc1ccco1)C(=O)Nc1ccc(Oc2ccc(Cl)cc2)nc1 | N#Cc1cccc1N1CCCN(C(=O)NCC(=O)N2CCCCC2)CC1 | Cc1cccc1N1CCCN(C(=O)NCC(=O)N2CCCCC2)CC1 |
| (d) Task: Less like a drug | | (e) Task: Higher permeability | | (f) Task: Lower permeability | |
| Input Molecule | Optimized Molecule | Input Molecule | Optimized Molecule | Input Molecule | Optimized Molecule |
| CCNC(=O)c1ccc(C)c(NC(=O)CCCO)c1 | CCNC(=O)c1ccc(C)(NC(=O)CCCOC(=O)O)c1 | O=C(NCC1CC1)c1csc(-c2cccc(OCc3cccc(F)c3)c2)n1 | O=C(NCC1CC1)c1csc(-c2cccc(C)c2)n1 | NCc1ccc(C(=O)Nc2ccc(Cl)c(S(=O)(=O)N3CCCC3)c2)cc1 | NCc1ccc(C(=O)Nc2ccc(Cl)c(S(=O)(=O)N3CCC(O)C3)c2)cc1 |

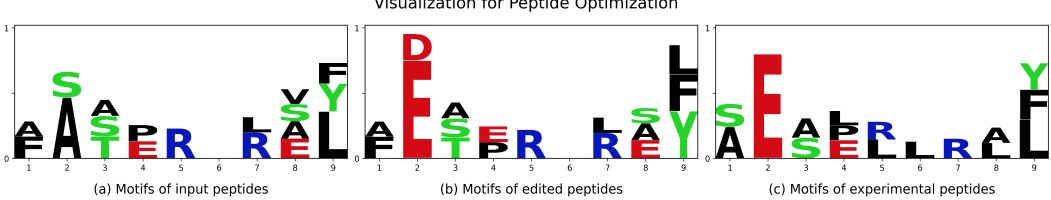

Figure 2: Visualization of peptide optimization with ChatGPT-4.1 for the task of optimizing peptides that bind to HLA-C16:01 into peptides that bind to HLA-B44:02. We provide the visualization using position weight matrices (PWMs). PWMs have been widely used for the visualization of protein motifs (patterns), and they plot the distribution of each amino acid at the corresponding position. Thus, more important motifs with higher probabilities will be marked with higher letters.

Table 14: Visualization of four protein optimization tasks. The input protein is shown in light grey, and the final optimized protein is shown in red.

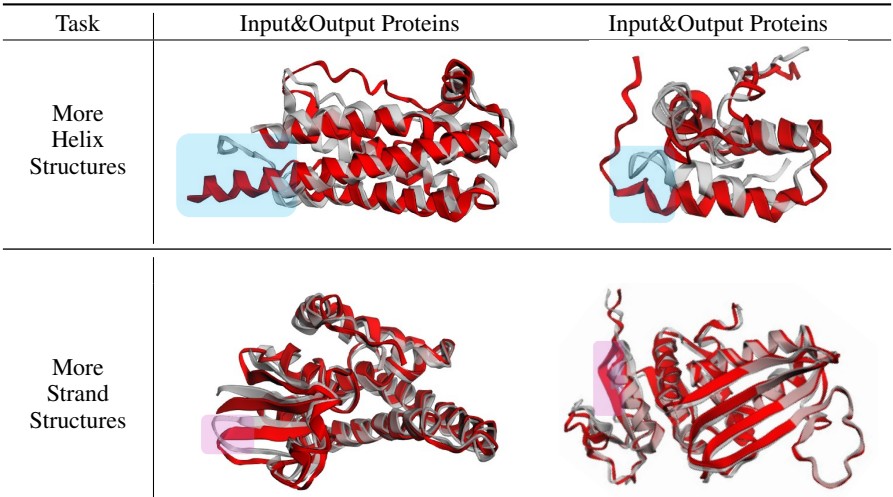

| Task | Input&Output Proteins | Input&Output Proteins |
|---|---|---|
| More Helix Structures | | |
| More Strand Structures | | |