# OpenReview forum: "Feedback to Reasoning: LLM-Assisted Molecular Optimization with Domain Feedback and Historical Reasoning"
_ICLR.cc/2026/Conference — ICLR 2026 Conference Withdrawn Submission_

### Official Review · Reviewer_9atv · 2025-10-27

**Soundness:** 2
**Presentation:** 3
**Contribution:** 2
**Rating:** 4
**Confidence:** 3

**Summary:**

This submission focuses on the molecular optimization problem, which requires iterative refinement based on domain-specific feedback on the proposal. The form of iterative improvement aligns the key strengths of LLMs, their capacity for interactive dialogue, and incorporates feedback. However, prior LLM-based methods overlook the importance of historical knowledge accumulation during refinement and utilize static, single-action suggestions from a limited action space. To address this limitation, the submission proposes feedback to reasoning (F2R), an agent framework that cooperatively guided self-reflection, domain-specific feedback, and memory mechanisms to enhance the quality of the proposed molecules. The submission conducts extensive empirical studies and justifies the effectiveness of the proposed method.

**Strengths:**

- The submission focuses on the trendy LLM reasoning problem. The motivation is clear: utilizing historical information is crucial for the iterative optimization tasks.
- The submission is generally well-written, with clear illustrations and tables.
- The conducted experiments provide empirical evidence on the effectiveness of the proposed method compared to the baselines.

**Weaknesses:**

- The submission positions F2R as the first framework to incorporate historical feedback and reasoning trajectories. However, such a memory and feedback mechanism already appears in prior agentic frameworks like OctoTools [1] and LangGraph. In addition, the F2R's memory structure seems a direct adaptation of existing agentic memory paradigms, adapting to the molecular domain with detailed prompt design and searching metric.
- The submission provides a limited ablation study on the proposed method. There is no quantitative breakdown of how much each component, including feedback, SAR knowledge, or memory, contributes to the overall performance. In addition, the submission does not examine the influence of memory size, retrieval count during reasoning, or similarity metrics, which are critical to validate claims of “dynamic knowledge accumulation.”
- Despite repeatedly using terms like “guided self-reflection”, the F2R does not demonstrate actual reasoning improvement over time. There is no evidence that the model’s reasoning quality changes or that stored reasoning traces meaningfully influence future behavior.

[1] OctoTools: An Agentic Framework with Extensible Tools for Complex Reasoning. In arXiv, 2025.

**Questions:**

1. Could you compare and discuss the model's reasoning behavior when cooperating with the F2R and other frameworks?
2. Could you provide ablation studies on the F2R's components, including feedback, SAR knowledge, and memory?
3. Could you discuss the difference between F2R and other agent frameworks with the memory component and the feedback mechanism?

---

> ### Author Response · Authors · 2025-11-18
> **Responses to Reviewer 9atv - Part I**
>
> We sincerely appreciate your valuable feedback. We have revised the manuscript and provide our detailed responses below.
>
> ### Weaknesses
> - W1: Comparison with prior agentic frameworks with memory and feedback mechanisms
>     - **We emphasize that we are the first to adopt and tailor these mechanisms for molecular optimization. Specifically, we employ domain-specific similarity metrics for retrieval, summarize domain-specific SAR patterns, and incorporate domain tools to provide meaningful feedback.** We believe these innovations are novel in LLM-assisted molecular optimization. With these innovations, we are able to demonstrate that LLMs can perform molecular optimization remarkably well, achieving success rates above 90% on many tasks.
>     - We acknowledge that memory and feedback mechanisms have been established in prior works and applied in other domains. We do not claim general novelty of these mechanisms across all domains. **We have included a discussion under Sec. 4 to discuss prior agentic frameworks, including OctoTools, and to better clarify our contributions.**
>
> - W2: Ablation study on the contribution of each component and on the influence of memory size and retrieval count during reasoning
>     - **We already have an ablation study in the original submission (Appendix D) that separates the contributions of memory and feedback (self-reflection).**
>         - **Specifically, in this ablation study, we show that both feedback-driven reasoning and knowledge accumulation are crucial components of LLM-assisted molecular optimization.** For example, in one task, providing history alone improves the success rate from 80.50% to 86.50%, while providing feedback alone improves it from 80.50% to 94.00%. When both are combined, the success rate reaches 99%.
>     - **In the revised paper, we have included an additional ablation study in Appendix D.2 analyzing how memory size and retrieval count affect performance.**
>         - **Specifically, the results demonstrate that increasing the amount of historical knowledge yields clear gains in success rate, providing direct evidence for the benefits of knowledge accumulation.** For example, the success rate increases from 80.50% to 84.00%, 86.00%, and 87.50% as the memory size grows from 0 to 100, 200, and 500, respectively.
>     - **In addition, in the revised paper, we have included a failure analysis in Appendix D.3 to provide the reasons of failure for each method, to offer another perspective on the sources of improvement achieved by F2R.** In this analysis, we can clearly see that feedback and memory help greatly with both structural validity and satisfactory propert values.
>
>
>
> - W3: On guided self-reflection
>     - **There is a misunderstanding.** By guided self-reflection, we refer to the feedback mechanism within each individual optimization task, not to memory shared across different tasks. In Sec. 4.1, we state "$\gamma_{i-1}$ is the self-reflection on the last failed result $x_{i-1}$ based on the feedback $f_{i-1}$." This clearly clarifies what we mean by self-reflection. However, we recognize that the confusion may stem from how our first contribution was originally phrased, and **we have reworded this contribution at the end of the introduction to avoid any ambiguity in the revised paper.**
>     - **Regarding how the model’s reasoning quality changes with such self-reflection/feedback, our ablation study (see Sec. 5.4, with results in Appendix D.2 due to space constraints) shows that using F2R-Feedback alone can lead to a substantial improvement in success rate (from 80.50% to 94.00% for one task and 74.50% to 91.50% for another).** This clearly demonstrate that self-reflection improves the reasoning capability and improves the overall performance.

---

> > ### Author Response · Authors · 2025-11-18
> > **Responses to Reviewer 9atv - Part II**
> >
> > ### Questions
> >
> > - Q1: Case study of reasoning behavior
> >     - **We are not sure we fully understand your question.** Do you mean comparing the reasoning behavior of F2R with *“the model’s reasoning behavior when cooperating with F2R and other frameworks”*? By “other frameworks,” are you referring to memory- and feedback-based systems such as OctoTools and LangGraph?
> >     - If we understand your question correctly, we would say F2R and these frameworks can both provide memory and feedback capabilities. **However, an important distinction is that F2R is specifically designed for LLM-assisted molecular optimization and includes components tailored to this domain that these more general frameworks do not provide.** Therefore, we don't think F2R and these frameworks are comparable.
> >     - **Nevertheless, we have provided case studies of the reasoning behaviors of F2R equipped with the feedback module and the historical knowledge module in Appendix D.4 in the revised paper**, to illustrate the LLM-assisted molecule optimization process and how feedback-driven self-reflection and historical knowledge lead to successful optimization results.
> >         - **Specifically, we provide an example in which the feedback enables the LLM to correct its own mistake and produce a valid, desired molecule.** In contrast, we also provide an example, with the same task and the same first round result, without such feedback, the LLM repeats the same mistake and generates an invalid molecule.
> >         - **We also present two cases showing the role of historical knowledge in optimization: an example where the model benefits from prior successes, and another example where it learns from past failures to avoid repeating the same errors.**
> >
> >
> > - Q2: Ablation study
> >     - **We have already provided this ablation study in the original submission. Please refer to W2 for further details.**
> >
> > - Q3: memory component and the feedback mechanism?
> >     - **We have added a brief discussion under Sec. 4 to clarify that F2R contains explicit domain-specific components and that we do not claim any general novelty regarding memory or feedback mechanisms across all domains.**

---

> > > ### Comment · Reviewer_9atv · 2025-11-19
> > >
> > > I thank the authors for their detailed responses to my questions and for addressing all my concerns. I have raised my score accordingly (from 4 to 6).
> > >
> > > W1-W3 and Q2-Q3: Thank you for the additional discussion and experiments.
> > >
> > > Q1: Yes, you understand my problem correctly. By reasoning behavior, I mean whether the model can conduct self-refinement or self-reflection based on the previous attempts.
> > >
> > > A follow-up question is: are these mechanisms (memory and feedback) also applicable to the small and open-sourced LLMs such as Qwen-2.5-7B Instruct or even Llama-3.2-3B Instruct? Can these models exhibit similar self-correction behavior to ChatGPT-4.1 using the F2R framework?

---

> ### Author Response · Authors · 2025-11-22
> **Second Round Responses to Reviewer 9atv - Part I**
>
> Thank you for your response. **We are pleased that all previously raised concerns have been satisfactorily resolved. Please find our answer to your new question below:**
>
> - Applicability to small models
>     - **The framework itself is independent of the specific model used, as long as the model can interpret the feedback and past experiences. However, its overall effectiveness is defintely correlated with the model’s capacity for understanding and reasoning about molecular optimization tasks.**
>     - **We evaluated F2R on *Meta-Llama-3.1-8B-Instruct* and *Qwen2.5-7B-Instruct*.** The 3B variant exhibits extremely limited chemical reasoning ability and is unable to effectively reason about chemical structures.
>     - **We provide case studies on these two models below. Overall, similar self-correction behaviors can be observed.** However, we must be upfront that we find it less effective than GPT-4.1, both with and without guidance, likely due to the limited reasoning capacity and chemical knowledge of these smaller models [1].
>
>     [1] Developing ChemDFM as a large language foundation model for chemistry, Cell Report Physical Science
>
> -   **Note:** `In all examples below, the sentences marked in this color (e.g. This is an invalid molecule due to...) are not LLM outputs; they are added to help readers better understand the results.`
>
>     - **Example 1 — Successful Self-Correction With Feedback** [Meta-Llama-3.1-8B-Instruct]:
>
>         **Input Molecule:** *CCCCn1nc(C(=O)Nc2ccc(N3CCOCC3)nc2)ccc1=O*
>
>         **Task:** *More soluble in water*
>
>         **Optimized Molecule (1st iteration):** *CCCC(=O)Nc1ccc(N2CCOCC(=O)C2)n1*  `(This is an invalid molecule due to a syntax error that RDKit cannot assign alternating single and double bonds to the aromatic atoms while keeping all valences valid)`
>
>         **Feedback:** *The generated molecule CCCC(=O)Nc1ccc(N2CCOCC(=O)C2)n1 is checked by experts and it is NOT valid because of RDKit Error: cannot kekulize molecule (atoms: 6, 7, 8, 9, 18). This means RDKit cannot assign alternating single and double bonds to the aromatic atoms while keeping all valences valid. Can you fix the issue?*
>
>         **Reasoning:**  *The error occurs because the ring system in the molecule is not able to be kekulized, which means that RDKit cannot correctly assign double bonds and single bonds to the aromatic ring. To fix the issue, I'll modify the molecule to ensure that it can be kekulized. Here's a revised version of the molecule:
>         CC(=O)Nc1ccc(N2CCOCC(=O)C2)nc1
>         I removed one of the carbon atoms from the chain to prevent the ring system from becoming too complex and to allow RDKit to correctly kekulize it.*
>
>
>         **Optimized Molecule (2nd iteration):** *CC(=O)Nc1ccc(N2CCOCC(=O)C2)nc1* `(This is a valid molecule that is more soluble in water)`
>
>     - **Example 2 - Failure Without Feedback** (using the same first-round messages as input to control for variables) [Meta-Llama-3.1-8B-Instruct]:
>
>         **Input Molecule:** *CCCCn1nc(C(=O)Nc2ccc(N3CCOCC3)nc2)ccc1=O*
>
>         **Task:** *More soluble in water*
>
>         **Optimized Molecule (1st iteration):** *CCCC(=O)Nc1ccc(N2CCOCC(=O)C2)n1*  `(This is an invalid molecule due to a syntax error that RDKit cannot assign alternating single and double bonds to the aromatic atoms while keeping all valences valid)`
>
>         **Feedback:** *The generated molecule CCCC(=O)Nc1ccc(N2CCOCC(=O)C2)n1 is checked by experts and it is NOT valid because of RDKit Error: cannot kekulize molecule (atoms: 6, 7, 8, 9, 18). This means RDKit cannot assign alternating single and double bonds to the aromatic atoms while keeping all valences valid. Can you fix the issue?*
>
>         **Reasoning:**  *To fix the molecule CCCC(=O)Nc1ccc(N2CCOCC(=O)C2)n1, I reinsert the nitrogen atom in the correct position and add the rest of the structure. Here's the corrected molecule: C(=O)Nc1ccc(n2ccccc2cn3ccccc3)NC(=O)c1.*
>
>
>         **Optimized Molecule (2nd iteration):** *C(=O)Nc1ccc(n2ccccc2cn3ccccc3)NC(=O)c1* `(This is still invalid for the same reason: the model assumes the structure is incorrect because it is not sufficiently soluble in water, so it adds the nitrogen atom back to increase solubility)`

---

> ### Author Response · Authors · 2025-11-22
> **Second Round Responses to Reviewer 9atv - Part II**
>
> - With Qwen2.5-7B-Instruct
>     - **Similar behavior can also be observed with Qwen2.5-7B-Instruct. We provide a successful case below for illustration:**
>
>         **Input Molecule:** *CCCCn1nc(C(=O)Nc2ccc(N3CCOCC3)nc2)ccc1=O*
>
>         **Task:** *More soluble in water*
>
>         **Optimized Molecule (1st iteration):** *CCCCn1nc(C(=O)Nc2ccc(N3CCOCC3)nc2)c1O*  `(This is an invalid molecule due to a syntax error that RDKit cannot assign alternating single and double bonds to the aromatic atoms while keeping all valences valid)`
>
>         **Feedback:** *The generated molecule CCCCn1nc(C(=O)Nc2ccc(N3CCOCC3)nc2)c1O is checked by experts and it is NOT valid because of RDKit Error: cannot kekulize molecule (atoms: 5, 6, 22). This means RDKit cannot assign alternating single and double bonds to the aromatic atoms while keeping all valences valid. Can you fix only this one?*
>
>         **Reasoning:**  *Certainly! The issue with the molecule \`CCCCn1nc(C(=O)Nc2ccc(N3CCOCC3)nc2)c1O\` is likely due to the difficulty in kekulizing the molecule, particularly around the aromatic ring system. We can address this by modifying the structure to ensure it can be kekulized properly.
>         Here is a modified version of the molecule that should be valid and more soluble in water:
>         \- \*\*CCCCn1nc(C(=O)Nc2ccc(N3CCOCC3)nc2)nc1O\*\*
>         \#\#\# Reasoning:
>         \- \*\*Modification of the Ring Structure:\*\* By changing the structure slightly, we can ensure that the aromatic ring can be kekulized properly. The key is to maintain the necessary functional groups while ensuring the molecule is still hydrophilic.
>         \#\#\# SMILES Representation:
>         \- \*\*CCCCn1nc(C(=O)Nc2ccc(N3CCOCC3)nc2)nc1O\*\*
>         This modification should resolve the kekulization issue while maintaining the desired hydrophilic properties.*
>
>         `(In the above, Qwen produces headings, bullet points, etc. in Markdown, and we show its output directly; "### Reasoning:" is also part of Qwen's response)`
>
>         **Optimized Molecule (2nd iteration):** *CCCCn1nc(C(=O)Nc2ccc(N3CCOCC3)nc2)nc1O* `(This is a valid molecule that is more soluble in water)`

---

> > ### Comment · Reviewer_9atv · 2025-11-27
> >
> > Thank you for the case studies on additional LLMs. I have no further comments.

---

### Official Review · Reviewer_XS5Z · 2025-10-30

**Soundness:** 2
**Presentation:** 2
**Contribution:** 2
**Rating:** 4
**Confidence:** 4

**Summary:**

This work introduces a molecular optimization pipeline based on LLMs, utilizing domain feedback and summarizing historical knowledge to iteratively optimize results. The manuscript is well-organized. The proposed pipeline exceeds baselines in both small molecule and peptide optimization tasks. As for its novelty, I cannot precisely assess its originality, but the pipeline seems rather straightforward and may lack technical depth, potentially making it better suited for conferences like *CL Findings.

Regarding experimental results, although the proposed F2R achieves impressive performance, such as a 99% success rate in molecular solubility optimization, there are notable deficiencies in key metrics and analyses. For instance, it is crucial to examine the similarity between optimized and original molecules, as well as the overall similarity among optimized molecules, given the use of historical knowledge which may result in the generation of identical or very similar molecules. Additionally, the diversity of optimized molecules should be assessed for the same reason. Visualizing some pre- and post-optimization pairs would also be highly beneficial. Moreover, examples where certain LLMs initially made incorrect inferences but corrected them based on provided feedback are crucial. Based on my experience, even models like GPT-4.1 encounter challenges in adjusting reasoning appropriately to align with detailed domain-specific feedback.

**Strengths:**

This work introduces a molecular optimization pipeline based on LLMs, utilizing domain feedback and summarizing historical knowledge to iteratively optimize results. The manuscript is well-organized.

**Weaknesses:**

The pipeline seems rather straightforward and may lack technical depth. Deficiencies in metrics and analyses.  For instance, it is crucial to examine the similarity between optimized and original molecules, as well as the overall similarity among optimized molecules, given the use of historical knowledge which may result in the generation of identical or very similar molecules. Additionally, the diversity of optimized molecules should be assessed for the same reason. Visualizing some pre- and post-optimization pairs would also be highly beneficial. Moreover, examples where certain LLMs initially made incorrect inferences but corrected them based on provided feedback are crucial.

**Questions:**

see Weaknesses

---

> ### Author Response · Authors · 2025-11-18
> **Responses to Reviewer XS5Z - Part I**
>
> Thank you for the helpful feedback. We have made revisions to the paper and respond to your comments below. For clarity, we have organized your comments into groups.
>
> ### Weaknesses
> - W1: Technical depth
>     - **We emphasize that we are the first to adopt and tailor momery and feedback mechanisms for LLM-assisted molecular optimization.** Specifically, we employ domain-specific similarity metrics for retrieval, summarize domain-specific SAR patterns, and incorporate domain tools to provide meaningful feedback. **We discuss how these innovations are novel and important in LLM-assisted molecular optimization below:**
>         - **First, we retrieve the most relevant past editing episodes using domain-specific similarity measures developed for biological sequences, rather than relying on the cosine similarity commonly used in NLP.**
>         - **Next, we summarize SAR patterns, transforming raw editing histories into structured, domain-relevant optimization heuristics.** These SAR patterns capture how specific structural modifications influence property values, biological activity, or binding affinity under a given optimization objective. **Unlike common summarization tasks in NLP, which focus on textual abstraction or theme extraction, SAR pattern summarization is interpreting chemistry- or biology-specific transformations**.
>         -  **Last but not least, we provide domain feedback to the LLM.** Feedback is crucial for enabling the LLM to understand its own mistakes and to self-reflect. We provide an illustrative example in the case studies in Appendix D.4 of the revised paper. **In this example, we clearly show that the feedback enables the LLM to correct its own mistake and produce a valid, desired molecule. In contrast, we also provide an example, with the same task and the same first round result, without such feedback, the LLM repeats the same mistake and generates an invalid molecule.**
>     - **Only with these innovations, we are able to demonstrate, for the first time, that LLMs can perform molecular optimization remarkably well, achieving success rates above 90% on many tasks.**
>
> - W2: Deficiencies in metrics and analyses; identical or very similar molecules
>     - **We follow exactly the same evaluation metrics and setup used in prior works [1, 2], which only accounts for validity and property values.** The success rate accounts for both the targeted property values and the validity of the generated molecules. **There might be a misunderstanding.**
>         - **For all optimization tasks in Tables 2, 3, 4, and 5, generating the same molecule as the input is counted as a failure, since it does not improve the desired property.**
>         - **If your concern refers to identical or very similar molecules appearing in the knowledge database, we offer the following clarification:** LLMs do not know the exact property values of either the input molecule or those in the knowledge database, and the molecules retrieved from past experience do not necessarily satisfy the optimization requirement. Therefore, the LLM cannot simply “look up” a molecule that meets the criteria.
>     - In addition, we also provide some visualizations as you suggest. In Tables 3 and 7 in the revised paper, we provide cases in which historical information helps the optimization process. **These examples clearly show that the LLM learns to use the prior knowledge and patterns rather than merely copying molecules from the knowledge base.** In Figure 3, the LLM demonstrates an understanding that incorporating a fluorine into the aromatic system is an effective strategy for improving drug-likeness, rather than merely replicating a molecule from its history.
>
> - W3: diversity of optimized molecules
>     - **We follow exactly the same evaluation metrics and setup used in prior works [1, 2].**
>     - **Regarding diversity, there might be a misunderstanding.**
>         - We have many different input molecules, and for each input molecule, the optimization is performed **independently**. For each case, we only need one successful candidate that satisfies the required property constraints. Because of this setup, diversity is typically not used as a evaluation metric for molecular optimization. **We are not sure whether we are misunderstanding your comments or whether there is a misunderstanding regarding our experimental setup.** We would like you to further clarify your concern.

---

> ### Author Response · Authors · 2025-11-18
> **Responses to Reviewer XS5Z - Part II**
>
> - W4.1 : Visualizing some pre- and post-optimization pairs
>     - **We have provided visualizations of optimized molecules in the revised paper.** **Specifically, in Table 3 under Sec. 5.1 and Table 7 under Sec. 5.3 of the revised paper**. In these visualizations, we provide the pre- and post-optimization molecule pairs along with the relevant historical examples retrieved from the historical knowledge base that support the optimization process.
>     - **We also provide additional general visualizations for all three types of optimization tasks (small molecules, immunogenic binding peptides, and protein secondary structures) in Appendix D.5.** These examples clearly show how F2R applies structured reasoning and produces improved candidates with feedback and memory. It also demonstrates the effectiveness and underlying mechanisms of LLM-assisted molecular optimization, particularly with feedback and history.
> - W4.2: Case studies of correction by feedback
>     - **We have provided case studies in Appendix D.4 in the revised paper** to illustrate the LLM-assisted molecule optimization process and how feedback-driven self-reflection and historical knowledge lead to successful optimization results.
>     - **Specifically, we provide an example in which the feedback enables the LLM to correct its own mistake and produce a valid, desired molecule.** In contrast, we also provide an example, with the same task and the same first round result, **without** such feedback, the LLM repeats the same mistake and generates an invalid molecule. **We also present two cases showing the role of historical knowledge in optimization: an example where the model benefits from prior successes, and another example where it learns from past failures to avoid repeating the same errors.**
>     - **For your convenience, we present two examples below (they are also in Appendix D.4)**: one illustrating the role of feedback and the other illustrating the role of historical information.
>
> -   **Note:** `In all examples below, the sentences marked in this color (e.g. This is an invalid molecule due to...) are not LLM outputs; they are added to help readers better understand the results.`
>     - Example 1 (Feedback):
>
>         **Input Molecule:** *Cn1ccc(C(=O)Nc2sc3c(c2C#N)CCC3)cc1=O*
>
>         **Task:** *More soluble in water*
>
>         **Optimized Molecule (1st iteration):** *HOCH2c1cc(Sc2cc3ccccc3[nH]2)ccc1C=O*  `(This is an invalid molecule due to a syntax error in the methylene group)`
>
>         **Feedback:** *The generated drug HOCH2c1cc(Sc2cc3ccccc3[nH]2)ccc1C=O is checked by experts and it is NOT valid because of RDKit error: ``syntax error while parsing; check for mistakes around position 1; failed to parse SMILES.''*
>
>         **Reasoning:**  *The previous SMILES HOCH2c1cc(...) was syntactically incorrect for a hydroxymethyl group. The corrected SMILES replaces the methyl group with a hydroxymethyl group (CH2OH) by explicitly defining the bond from the ring carbon to the new carbon, which is then bonded to oxygen, thereby increasing water solubility through the addition of a hydroxyl group.*
>
>         **Optimized Molecule (2nd iteration):** *c1(CO)cc(Sc2cc3ccccc3[nH]2)ccc1C=O}* `(This is a valid molecule that is more soluble in water)`
>     - Example 2 (History):
>
>         **Input Peptide:** *ITAPTVRQY*
>
>         **Task:** *We want a peptide that binds to HLA-B\*44:02. We have a peptide ITAPTVRQY that binds to HLA-C\*16:01, can you help modify it?*
>
>         **Optimized Peptide (1st iteration):** *IPAPTVRQY* `(This is a valid peptide; however, it does not bind to HLA-B*44:02)`
>
>         **Historical Knowledge:**  *Original Peptide: AEAGVRLLSY; Optimized Peptide: AEAGVRLLSW; Summary: Substitution to W at the C-terminus enhanced binding to HLA-B\*44:02*
>
>         **SAR:**  *C-terminal tryptophan (W) substitution improves HLA-B\*44:02 binding*
>
>         **Reasoning:**  *The feedback from “History 2” indicated that substituting the C-terminal Tyrosine (Y) with Tryptophan (W) significantly enhanced binding to HLA-B\*44:02 (AEAGVRLLSY → AEAGVRLLSW). Given that the current peptide ITAPTVRQY also has a Tyrosine at its C-terminus, replacing it with Tryptophan is a direct application of a previously successful modification to improve HLA-B\*44:02 binding.*
>
>         **Optimized Peptide (2nd iteration):** *ITAPTVRQW*
>           `(This is a valid peptide, and it binds to HLA-B*44:02)`
>
> [1] Conversational Drug Editing Using Retrieval and Domain Feedback, ICLR 2024
>
> [2] RL-Guider: Leveraging Historical Decisions and Feedback for Drug Editing with Large Language Models, ACL Findings 2025

---

> > ### Author Response · Authors · 2025-11-25
> > **Follow-up on Our Reponses - Reviewer XS5Z**
> >
> > Dear Reviewer XS5Z,
> > - **It has been almost a week since we posted our rebuttal responses, so we would like to follow up to see if you have any remaining concerns.**
> > - Specifically, we have taken the following steps to address your comments:
> >     1. We have provided additional explanation of our contributions.
> >     2. We have clarified that our evaluation metrics follow prior work exactly, and we further addressed the concern regarding similarity and diversity by adding visualizations and case studies.
> >     3. We have included pre- and post-optimization molecule pairs, as well as case studies illustrating the actual LLM-assisted optimization process with F2R.
> >
> > **Please let us know if you have any remaining concerns; otherwise, we would greatly appreciate it if you could reconsider your rating.**

---

> > > ### Comment · Reviewer_XS5Z · 2025-11-27
> > >
> > > Thank you for the author's response and efforts. I believe the revised version is a nice report, but it may not meet the standards of ICLR. For those who need to use LLMs for molecular optimization, memory mechanisms and feedback mechanisms are tools they use, consciously or unconsciously, on a daily basis. For most LLM service providers, conversation caching is a basic feature; whereas interactive feedback happens spontaneously during the use of LLMs. Regarding the optimization of small or large molecules, I do not see from this work the capabilities and limitations of LLMs—such as which tasks are suitable and which are not. Additionally, this work is heavily dependent on ChatGPT and Gemini, two models that may not be easily accessible in certain regions. Therefore, I consider this work to offer a good report rather than a highly insightful research contribution. It might be more suitable for publication at *CL Findings or similar events.

---

> > > > ### Author Response · Authors · 2025-11-27
> > > > **Second Round Responses to Reviewer XS5Z - Part I**
> > > >
> > > > Thank you for your response and for your willingness to engage in discussion. We deeply appreciate and respect your feedback; however, we would like to offer a few clarifications.
> > > >
> > > >
> > > > - "insightful research contribution" of memory and feedback
> > > >     - This is a subjective; different people might have different opinions, and **we totally understand and respect your opinion**.
> > > >     - However, **we would like to clarify that our mechanisms are different from that offered by LLM service providers.**
> > > >     - **In short: (1) The memory mechanisms provided by LLM service providers cannot faithfully store or retrieve historical molecular optimization results due to the lack of domain-specific considerations; and (2) The spontaneous interactive feedback you referred to requires human intervention, whereas our focus is on a fully autonomous pipeline.**
> > > >     - **Detailed response:**
> > > >         - The memory mechanism provided by LLM service providers is rather general. After several optimization tasks, the conversation cache becomes extremely long and must be summarized. Even then, the amount of historical context that can be stored is limited. However, in drug discovery tasks, the number of molecular optimization steps required for a single drug development project is numerous [1]. In addition, LLMs do not know how to accurately retrieve the most relevant past information based on NLP similarity metrics, e.g. cosine similarity based on embeddings. **In contrast, we can store all past optimization results and extract structure–activity relationship (SAR) patterns. We then retrieve the most relevant results using chemically meaningful metrics based on the input molecules. Only in this way, historical knowledge can be accurately accumulated.**
> > > >         - For the feedback mechanism, we agree that “interactive feedback happens spontaneously during the use of LLMs.” However, in drug discovery tasks, the number of molecular optimization steps required for a single drug development project is numerous [1]. The spontaneous interactive feedback you refer to relies on human involvement, whereas our approach emphasizes automation driven by domain-specific software. **As we clarify in Sec. 4.1 under “Autonomous Domain Feedback,” our work is focused on building a fully autonomous pipeline. For similar reasons, none of the existing works incorporate feedback mechanisms at all since spontaneous interactive feedback requires human intervention.**
> > > >
> > > >
> > > > - I do not see from this work the capabilities and limitations of LLMs—such as which tasks are suitable and which are not.
> > > >     - **Our benchmarking tasks are exactly the same as those in prior works, e.g., [2, 3].** LLMs have already demonstrated strong capabilities in performing molecular optimization, and **our aim is to push the boundaries of LLM-assisted molecular optimization even further.** Tasks for which LLMs are not well suited are beyond the scope of this work.
> > > >     - In addition, our results show that, with domain-specific feedback and memory, LLMs can perform extremely well on the tasks established by [2]. **In fact, we are the first to demonstrate that LLMs can now even saturate performance (even over 95% success rates  for some tasks with F2R) on certain tasks, allowing future work to focus only on those tasks that remain genuinely challenging as a benchmark.** We have included a discussion of this at the end of Sec. 5.3 in the revised paper. In this discussion, we also advocate for a reduced version of the evaluation benchmark by removing tasks in which F2R has already demonstrated extremely high success rates. Details are provided in Appendix D.1.
> > > >
> > > >
> > > > - Additionally, this work is heavily dependent on ChatGPT and Gemini, two models that may not be easily accessible in certain regions.
> > > >     - **Most LLM models with strong knowledge bases and reasoning capabilities are not easily accessible**, as they either require powerful GPUs to host or incur monetary costs when accessed through an API.
> > > >     - **The task of drug discovery, for which molecular optimization is an essential subtask, costs money and time.** Drug discovery is a long and expensive process that developing a new drug takes up to 12-15 years and costs money on the order of billions [1,4,5]. **Therefore, if LLMs can help push the boundaries even a little, they are worth the cost.**
> > > >     - We understand your concern. **Our proposed mechanisms also work for smaller LLMs.** We will include several case studies (example executions) in a separate reply.
> > > >
> > > >
> > > >
> > > > [1] Drug discovery and development: introduction to the general public and patient groups
> > > >
> > > > [2] Conversational Drug Editing Using Retrieval and Domain Feedback
> > > >
> > > > [3] RL-Guider: Leveraging Historical Decisions and Feedback for Drug Editing with Large Language Models
> > > >
> > > > [4] Estimated Research and Development Investment Needed to Bring a New Medicine to Market, 2009-2018
> > > >
> > > > [5] How Much Does It Cost to Research and Develop a New Drug? A Systematic Review and Assessment

---

> > > > > ### Author Response · Authors · 2025-11-27
> > > > > **Second Round Responses to Reviewer XS5Z - Part II - Case Studies**
> > > > >
> > > > > - We provide examples using small models, Meta-Llama-3.1-8B-Instruct and Qwen2.5-7B-Instruct.
> > > > > -   **Note:** `In all examples below, the sentences marked in this color (e.g. This is an invalid molecule due to...) are not LLM outputs; they are added to help readers better understand the results.`
> > > > >
> > > > > - With Meta-Llama-3.1-8B-Instruct
> > > > >     - **Example 1 — Successful Self-Correction With Feedback** [Meta-Llama-3.1-8B-Instruct]:
> > > > >
> > > > >         **Input Molecule:** *CCCCn1nc(C(=O)Nc2ccc(N3CCOCC3)nc2)ccc1=O*
> > > > >
> > > > >         **Task:** *More soluble in water*
> > > > >
> > > > >         **Optimized Molecule (1st iteration):** *CCCC(=O)Nc1ccc(N2CCOCC(=O)C2)n1*  `(This is an invalid molecule due to a syntax error that RDKit cannot assign alternating single and double bonds to the aromatic atoms while keeping all valences valid)`
> > > > >
> > > > >         **Feedback:** *The generated molecule CCCC(=O)Nc1ccc(N2CCOCC(=O)C2)n1 is checked by experts and it is NOT valid because of RDKit Error: cannot kekulize molecule (atoms: 6, 7, 8, 9, 18). This means RDKit cannot assign alternating single and double bonds to the aromatic atoms while keeping all valences valid. Can you fix the issue?*
> > > > >
> > > > >         **Reasoning:**  *The error occurs because the ring system in the molecule is not able to be kekulized, which means that RDKit cannot correctly assign double bonds and single bonds to the aromatic ring. To fix the issue, I'll modify the molecule to ensure that it can be kekulized. Here's a revised version of the molecule:
> > > > >         CC(=O)Nc1ccc(N2CCOCC(=O)C2)nc1
> > > > >         I removed one of the carbon atoms from the chain to prevent the ring system from becoming too complex and to allow RDKit to correctly kekulize it.*
> > > > >
> > > > >
> > > > >         **Optimized Molecule (2nd iteration):** *CC(=O)Nc1ccc(N2CCOCC(=O)C2)nc1* `(This is a valid molecule that is more soluble in water)`
> > > > >
> > > > >     - **Example 2 - Failure Without Feedback** (using the same first-round messages as input to control for variables) [Meta-Llama-3.1-8B-Instruct]:
> > > > >
> > > > >         **Input Molecule:** *CCCCn1nc(C(=O)Nc2ccc(N3CCOCC3)nc2)ccc1=O*
> > > > >
> > > > >         **Task:** *More soluble in water*
> > > > >
> > > > >         **Optimized Molecule (1st iteration):** *CCCC(=O)Nc1ccc(N2CCOCC(=O)C2)n1*  `(This is an invalid molecule due to a syntax error that RDKit cannot assign alternating single and double bonds to the aromatic atoms while keeping all valences valid)`
> > > > >
> > > > >         **Feedback:** *The generated molecule CCCC(=O)Nc1ccc(N2CCOCC(=O)C2)n1 is checked by experts and it is NOT valid because of RDKit Error: cannot kekulize molecule (atoms: 6, 7, 8, 9, 18). This means RDKit cannot assign alternating single and double bonds to the aromatic atoms while keeping all valences valid. Can you fix the issue?*
> > > > >
> > > > >         **Reasoning:**  *To fix the molecule CCCC(=O)Nc1ccc(N2CCOCC(=O)C2)n1, I reinsert the nitrogen atom in the correct position and add the rest of the structure. Here's the corrected molecule: C(=O)Nc1ccc(n2ccccc2cn3ccccc3)NC(=O)c1.*
> > > > >
> > > > >
> > > > >         **Optimized Molecule (2nd iteration):** *C(=O)Nc1ccc(n2ccccc2cn3ccccc3)NC(=O)c1* `(This is still invalid for the same reason: the model assumes the structure is incorrect because it is not sufficiently soluble in water, so it adds the nitrogen atom back to increase solubility)`

---

> > > > > > ### Author Response · Authors · 2025-11-27
> > > > > > **Second Round Responses to Reviewer XS5Z - Part III - Case Studies**
> > > > > >
> > > > > > - With Qwen2.5-7B-Instruct
> > > > > >     - **Similar behavior can also be observed with Qwen2.5-7B-Instruct. We provide a successful case below for illustration:**
> > > > > >
> > > > > >         **Input Molecule:** *CCCCn1nc(C(=O)Nc2ccc(N3CCOCC3)nc2)ccc1=O*
> > > > > >
> > > > > >         **Task:** *More soluble in water*
> > > > > >
> > > > > >         **Optimized Molecule (1st iteration):** *CCCCn1nc(C(=O)Nc2ccc(N3CCOCC3)nc2)c1O*  `(This is an invalid molecule due to a syntax error that RDKit cannot assign alternating single and double bonds to the aromatic atoms while keeping all valences valid)`
> > > > > >
> > > > > >         **Feedback:** *The generated molecule CCCCn1nc(C(=O)Nc2ccc(N3CCOCC3)nc2)c1O is checked by experts and it is NOT valid because of RDKit Error: cannot kekulize molecule (atoms: 5, 6, 22). This means RDKit cannot assign alternating single and double bonds to the aromatic atoms while keeping all valences valid. Can you fix only this one?*
> > > > > >
> > > > > >         **Reasoning:**  *Certainly! The issue with the molecule \`CCCCn1nc(C(=O)Nc2ccc(N3CCOCC3)nc2)c1O\` is likely due to the difficulty in kekulizing the molecule, particularly around the aromatic ring system. We can address this by modifying the structure to ensure it can be kekulized properly.
> > > > > >         Here is a modified version of the molecule that should be valid and more soluble in water:
> > > > > >         \- \*\*CCCCn1nc(C(=O)Nc2ccc(N3CCOCC3)nc2)nc1O\*\*
> > > > > >         \#\#\# Reasoning:
> > > > > >         \- \*\*Modification of the Ring Structure:\*\* By changing the structure slightly, we can ensure that the aromatic ring can be kekulized properly. The key is to maintain the necessary functional groups while ensuring the molecule is still hydrophilic.
> > > > > >         \#\#\# SMILES Representation:
> > > > > >         \- \*\*CCCCn1nc(C(=O)Nc2ccc(N3CCOCC3)nc2)nc1O\*\*
> > > > > >         This modification should resolve the kekulization issue while maintaining the desired hydrophilic properties.*
> > > > > >
> > > > > >         `(In the above, Qwen produces headings, bullet points, etc. in Markdown, and we show its output directly; "### Reasoning:" is also part of Qwen's response)`
> > > > > >
> > > > > >         **Optimized Molecule (2nd iteration):** *CCCCn1nc(C(=O)Nc2ccc(N3CCOCC3)nc2)nc1O* `(This is a valid molecule that is more soluble in water)`

---

### Official Review · Reviewer_Ayw5 · 2025-10-31

**Soundness:** 3
**Presentation:** 3
**Contribution:** 3
**Rating:** 6
**Confidence:** 4

**Summary:**

The paper introduces F2R, a framework that enhances LLMs for molecular optimization by integrating iterative domain feedback and memory-based reasoning. Unlike existing models that perform one-shot edits, F2R allows the LLM to receive structured feedback and self-reflect before proposing the next molecule. It also accumulates structure-activity relationship patterns from past tasks to inform future reasoning, functioning like an experience-driven agent.

**Strengths:**

1. The paper introduces a feedback loop to enable self-correction.

2. The approach employs an agentic memory that extracts reusable structure-activity relationship insights.

3. The results show significant gains over ChatDrug, ChemReasoner, and RL-Guider.

4. The work can generalize to other scientific reasoning tasks such as peptides and proteins.

**Weaknesses:**

1. The framework heavily relies on LLMs' reasoning quality, which may not generalize to weaker models.

2. The feedback and retrieval loops increase computational cost.

3. Knowledge reuse depends on task similarity, so the performance on unseen targets is unclear.

**Questions:**

1. Are there observed cases where the model produces the right answer with wrong reasoning or plausible reasoning leading to an incorrect molecule? If so, how does the feedback loop address such inconsistencies? Have the authors performed any systematic error analysis to categorize types of reasoning failures or feedback misinterpretations across tasks? Could accumulated reasoning errors propagate in memory?

2. F2R currently optimizes a single target property at a time, using feedback such as “increase solubility” or “improve binding affinity.” In realistic molecular design, however, optimization often involves multiple competing objectives. For example, we ask the model to simultaneously improve potency, solubility, and stability, or to balance drug-likeness with binding affinity.

How does F2R scale to such multi-property or high-dimensional optimization settings, where feedback must capture trade-offs between properties rather than a single scalar goal?

---

> ### Author Response · Authors · 2025-11-18
> **Responses to Reviewer Ayw5 - Part I**
>
> Thank you very much for your insightful feedback! We have revised the paper accordingly and provide our responses below.
>
> ### Weaknesses
> - W1: The framework heavily relies on LLMs' reasoning quality
>     - **This framework itself is independent of the LLMs’ reasoning quality. Rather than viewing the LLMs’ reasoning ability as a factor that weakens our proposed mechanism, we would argue that the model’s capability affects the entire LLM-assisted molecular optimization task.** If the underlying LLM is too weak, lacking chemical knowledge or reasoning ability, then the optimization task would fail regardless of the framework used. **Therefore, we kindly ask the reviewer to view this not as a limitation of our method, but as an inherent limitation of any LLM-assisted approach.**
> - W2: The feedback and retrieval loops increase computational cost
>     - **The feedback loop does not necessarily increase computational cost.** In each molecular optimization round, the optimized molecule must be evaluated anyway to determine whether further iterations are required; so there are natural feedback sources free of computational cost.
>     - **In fact, all forms of feedback used in this work impose minimal computational cost, as we do not observe any measurable cost in wall-clock runtime (evaluation feedback is free; RDKit feedback is extremely fast for a single molecule).** However, one can use feedback from advanced computational softwares, which may incur computational cost.
>     - **We agree that the retrieval process introduces some additional cost; however, this cost is negligible compared to the computational cost of the LLM itself.** For example, retrieving from a knowledge database of 200 optimization episodes for the task *“More soluble in water”* takes **less than 0.03 seconds**. Although we do not know the exact computation time of GPT-4.1 or Gemini-2.5-Flash since they are API-gated, it is clear that their inference time far exceeds that of retrieval given the same compute power.
>
> - W3: The performance on unseen targets
>     - This is a completely reasonable concern. **However, even when the knowledge base contains no similar past experience, or is entirely empty, the feedback mechanism remains highly effective, ensuring that the overall framework continues to work well.** We would say that the lack of data (or in our case past knowledge) is a general issue for most ML/DL algorithms, we would kindly ask the reviewer to not view this as a major weakness.
>         -  In Appendix D.2, we include an ablation study that separates the contributions of memory and feedback. Even the feedback mechanism alone provides a substantial improvement in success rates (e.g., from 80.50% to 94.00% for one task and from 74.50% to 91.50% for another).
>         -  **Therefore, even when the knowledge base contains no similar past experience, or is entirely empty, the feedback mechanism remains highly effective, ensuring that the overall framework continues to work well.**

---

> > ### Author Response · Authors · 2025-11-18
> > **Responses to Reviewer Ayw5 - Part II**
> >
> > ### Questions
> > - Q1.1: Correct reasoning leads to unsuccessful results, or incorrect reasoning leads to successful results; reasoning error propagation and accumulation
> >     - **This is a great question. Reasoning errors typically lead to unsuccessful results, and such failures can in fact help future optimization tasks avoid repeating the same mistakes.** However, in cases where correct reasoning leads to unsuccessful results, or incorrect reasoning leads to successful results, issues may arise. **We inspected a number of optimization episodes and did not observe such cases in our experiments. Nevertheless, we have included this point as a limitation and direction for future work in Section 6 of the revised paper.**
> > - Q1.2: Error analysis to categorize types
> >     - **We include a failure analysis in Appendix D.3 (first referenced in Section 5.4).**
> >     - For small molecules, the most common failures in baseline methods arise from invalid molecules (i.e., syntax failures in the SMILES strings). **F2R alleviates this issue by providing explicit feedback on these syntax failures.**
> >     - For peptides and proteins, such syntax failures naturally occur far less frequently; instead, the predominant failures involve property values that do not meet the task requirements. For roughly one third of the small-molecule optimization tasks, the failure is also due to unmet property constraints.
> >     - **In these cases, F2R provides detailed feedback on why the failure occurs (e.g., whether a property value is too high or too low, and for multi-objective tasks, which objectives are satisfied and which are not), along with guidance informed by historical optimization episodes.**
> >
> > - Q2: Multi-property setting
> >     - This is an excellent question, and **we already include multi-objective tasks in the original submission.** Across these tasks, F2R achieves the highest success rate in most cases and delivers substantial performance improvements
> >         - In Section 5.1, under the *Evaluation* paragraph, we state that we also evaluate the setting of ‘multi-objective optimization, where the goal is to improve two property values simultaneously.’ The corresponding results are presented in Table 3 in Section 5.1.
> >         - In Section 5.2, we have revised the paper to explicitly mention the presence of multi-objective tasks; this information was originally described in the appendix. The corresponding results are shown in Table 5 in Section 5.2.

---

### Official Review · Reviewer_RJqH · 2025-11-02

**Soundness:** 2
**Presentation:** 1
**Contribution:** 2
**Rating:** 2
**Confidence:** 4

**Summary:**

The paper proposes F2R, a framework for improving molecular design using feedback from experts in a loop, with a mechanism for self-reflection on mistaken actions. the framework leverages this to create historical feedback.
This feedback is automatically generated with another LLM. This historical feedback is then retrieved during inference-time, based on the similarity between the generated molecules.

**Strengths:**

The idea of using accumulated expert feedback to reason towards improving some objective is fundamentally attractive as it tackles a key problem of learning algorithms like RL, which is that "errors" and "successes" are not reflected upon, making the rewards very sparse.
The authors make a good case comparing with other similar works in the field like ChemReasoner and ChatDrug, and show that their system F2R performs the best on most tasks.

**Weaknesses:**

Code and data are not given for review.
many ablations missing
examples of execution would be great for understanding. right now it's not very transparent
results on the metric look suspiciously high, and there is no analysis of this. This seems to imply that the task has been technically solved, which is not the case. If the benchmarks have been saturated, the authors should either evaluate on a harder benchmark, or if not available select more advanced cases of the task, and see if the performance still holds. Potentially also adding some case-studies where you analyze how the system performs, what it gets correct and what it gets wrong, and how it compares against other methods.
Although the idea is intellectually appealing, the authors do not show any examples of how the feedback looks like, and if there are any specific insights that the system has leveraged towards completing the task.
It is a bit sad that a paper is written entirely about molecular generation and optimization, and yet no single depiction of a molecule is shown. I think it's important to show examples, to demonstrate what the solutions given by your system really look like, if they are any good. Simply showing a bunch of results on a table is not enough and at minimum lacks transparency.

**Questions:**

See the weaknesses.

---

> ### Author Response · Authors · 2025-11-18
> **Responses to Reviewer RJqH - Part I**
>
> Thank you very much for your valuable feedback! We have revised the paper and provide our responses below. For clarity, we have organized your comments into groups.
>
> ### Weaknesses
> - W1: Release of code and data
>     - **We understand your concern; we have cleaned and organized the code in this anonymous Git repository: [https://anonymous.4open.science/r/ICLR26-Submission-F2R-Repo-50B7/README.md](https://anonymous.4open.science/r/ICLR26-Submission-F2R-Repo-50B7/README.md).** In addition, we have included a README file with instructions on how to set up the environment, download the datasets and models, and run the code. You can verify yourself that F2R can achieve very high success rates on several tasks (e.g. 101: More soluble in water).
>     -  **Moreover, we want to emphasize that it is not required to release the code for submission per ICLR policy [1]; we hope that this should not be a weakness of this work.**
>     - **In the paper, we explicitly mention that we follow the exact same dataset and experimental setup used in the pioneering work ChatDrug [2].**
>         - In the first paragraph under Sec. 5.1: "The test molecules are sampled from the ZINC dataset, and following [2], the evaluation is conducted on a set of 200 molecules."
>         - In the first paragraph under Sec. 5.2: "We randomly pick 500 target-source pairs from 30 common MHC proteins (alleles) following [2] exactly."
>         - In the first paragraph under Sec. 5.3: "We use the test set of TAPE as our testing examples following [2]."
>
> - W2.1: Missing ablations
>     - **We already have an ablation study in the original submission (Appendix D) that separates the contributions of memory and feedback (self-reflection).**
>         - **Specifically, in this ablation study, we show that both feedback-driven reasoning and knowledge accumulation are crucial components of LLM-assisted molecular optimization.** For example, in one task, providing history alone improves the success rate from 80.50% to 86.50%, while providing feedback alone improves it from 80.50% to 94.00%. When both are combined, the success rate reaches 99%.
>     - **In the revised paper, we have included an additional ablation study in Appendix D.2 analyzing how memory size and retrieval count affect performance.**
>         - **Specifically, the results demonstrate that increasing the amount of historical knowledge yields clear gains in success rate, providing direct evidence for the benefits of knowledge accumulation.** For example, the success rate increases from 80.50% to 84.00%, 86.00%, and 87.50% as the memory size grows from 0 to 100, 200, and 500, respectively.
>     - **In addition, in the revised paper, we have included a failure analysis in Appendix D.3 to provide the reasons of failure for each method to offer another perspective on the sources of improvement achieved by F2R.** In this analysis, we can clearly see that feedback and memory help greatly with both structural validity and satisfactory propert values.

---

> ### Author Response · Authors · 2025-11-18
> **Responses to Reviewer RJqH - Part II**
>
> - W2.2: Missing Examples of Execution and Examples of Feedback
>     - **We have provided case studies in Appendix D.4 in the revised paper** to illustrate the LLM-assisted molecule optimization process and how feedback-driven self-reflection and historical knowledge lead to successful optimization results.
>         - **Specifically, we provide an example in which the feedback enables the LLM to correct its own mistake and produce a valid, desired molecule.** In contrast, we also provide an example, with the same task and the same first round result, **without** such feedback, the LLM repeats the same mistake and generates an invalid molecule.
>         - **We also present two cases showing the role of historical knowledge in optimization: an example where the model benefits from prior successes, and another example where it learns from past failures to avoid repeating the same errors.**
>     - **In the original paper, we already include an example of feedback at the end of Sec. 4.1. In the case studies mentioned above (Appendix D.4 of the revised paper), it is clearer what the feedback is and how it helps with the optimization task.**
>     - **For your convenience, we present two examples below (they are also in Appendix D.4)**: one illustrating the role of feedback and the other illustrating the role of historical information.
>
> -   **Note:** `In all examples below, the sentences marked in this color (e.g. This is an invalid molecule due to...) are not LLM outputs; they are added to help readers better understand the results.`
>     - Example 1 (Feedback):
>
>         **Input Molecule:** *Cn1ccc(C(=O)Nc2sc3c(c2C#N)CCC3)cc1=O*
>
>         **Task:** *More soluble in water*
>
>         **Optimized Molecule (1st iteration):** *HOCH2c1cc(Sc2cc3ccccc3[nH]2)ccc1C=O*  `(This is an invalid molecule due to a syntax error in the methylene group)`
>
>         **Feedback:** *The generated drug HOCH2c1cc(Sc2cc3ccccc3[nH]2)ccc1C=O is checked by experts and it is NOT valid because of RDKit error: ``syntax error while parsing; check for mistakes around position 1; failed to parse SMILES.''*
>
>         **Reasoning:**  *The previous SMILES HOCH2c1cc(...) was syntactically incorrect for a hydroxymethyl group. The corrected SMILES replaces the methyl group with a hydroxymethyl group (CH2OH) by explicitly defining the bond from the ring carbon to the new carbon, which is then bonded to oxygen, thereby increasing water solubility through the addition of a hydroxyl group.*
>
>         **Optimized Molecule (2nd iteration):** *c1(CO)cc(Sc2cc3ccccc3[nH]2)ccc1C=O}* `(This is a valid molecule that is more soluble in water)`
>     - Example 2 (History):
>
>         **Input Peptide:** *ITAPTVRQY*
>
>         **Task:** *We want a peptide that binds to HLA-B\*44:02. We have a peptide ITAPTVRQY that binds to HLA-C\*16:01, can you help modify it?*
>
>         **Optimized Peptide (1st iteration):** *IPAPTVRQY* `(This is a valid peptide; however, it does not bind to HLA-B*44:02)`
>
>         **Historical Knowledge:**  *Original Peptide: AEAGVRLLSY; Optimized Peptide: AEAGVRLLSW; Summary: Substitution to W at the C-terminus enhanced binding to HLA-B\*44:02*
>
>         **SAR:**  *C-terminal tryptophan (W) substitution improves HLA-B\*44:02 binding*
>
>         **Reasoning:**  *The feedback from “History 2” indicated that substituting the C-terminal Tyrosine (Y) with Tryptophan (W) significantly enhanced binding to HLA-B\*44:02 (AEAGVRLLSY → AEAGVRLLSW). Given that the current peptide ITAPTVRQY also has a Tyrosine at its C-terminus, replacing it with Tryptophan is a direct application of a previously successful modification to improve HLA-B\*44:02 binding.*
>
>         **Optimized Peptide (2nd iteration):** *ITAPTVRQW*
>           `(This is a valid peptide, and it binds to HLA-B*44:02)`

---

> > ### Author Response · Authors · 2025-11-18
> > **Responses to Reviewer RJqH - Part III**
> >
> > - W2.3: Missing visualization of optimized molecules
> >     - **We have provided visualizations of optimized molecules in the revised paper.** **Specifically, in Table 3 under Sec. 5.1 and Table 7 under Sec. 5.3 of the revised paper**.
> >         - In these visualizations, we provide the pre- and post-optimization molecule pairs along with the relevant historical examples retrieved from the historical knowledge base that support the optimization process.
> >     - **We also provide additional general visualizations for all three types of optimization tasks (small molecules, immunogenic binding peptides, and protein secondary structures) in Appendix D.5.**
> >         - These examples clearly show how F2R applies structured reasoning and produces improved candidates with feedback and memory. It also demonstrates the effectiveness and underlying mechanisms of LLM-assisted molecular optimization, particularly with feedback and history.
> >
> >
> > - W3: Choice of benchmarking dataset, analysis of "suspiciously high success rate"
> >     - **For the “suspiciously” high success rate on the test data, we emphasize that we follow the exact same test dataset and setup as in prior works [2, 3], as discussed in “Setup” under Sec. 5. The reason why the success rate is so high for some tasks can be attibuted to the improved reasoning and memory mechanisms of F2R.**
> >         - Specifically, providing domain feedback on the optimized candidates allows the LLM to recognize its own mistakes and make strategic adjustments during optimization (self-reflection). Equipping the LLM with a memory mechanism enables it to learn from past successes and failures, similar to human experts. In addition, the summarized structure–activity relationship from historical memory provides high-level actionable guidance that helps the LLM make more effective and chemically meaningful modifications. The effectiveness of our F2R mechanism is evident from the ablation studies and case analyses presented above in W2.
> >     - We agree that some tasks are fairly easy. **However, there are also many tasks on which all methods struggle.** For example, for most peptide and protein tasks, the success rate is below 90%, and for many, it is even below 70%. For multi-objective tasks, the success rates are often below 80%, and sometimes even below 40% (401, 402).
> >     - **In fact, we are the first to demonstrate that LLMs can now perform extremely well and even saturate performance on certain tasks, allowing future work to focus only on those tasks that remain genuinely challenging as a benchmark.** We have included a discussion of this at the end of Sec. 5.3 in the revised paper. In this discussion, we also advocate for a reduced version of the evaluation benchmark by removing tasks in which F2R has already demonstrated extremely high success rates. Details are provided in Appendix D.1.
> >
> >
> > [1] ICLR 2026 Author Guide: https://iclr.cc/Conferences/2026/AuthorGuide
> >
> > [2] Conversational Drug Editing Using Retrieval and Domain Feedback, ICLR 2024
> >
> > [3] RL-Guider: Leveraging Historical Decisions and Feedback for Drug Editing with Large Language Models, ACL Findings 2025
> >
> > ### Grouping of Weaknesses
> > Code and data are not given for review (W1). many ablations missing examples of execution would be great for understanding (W2). right now it's not very transparent results on the metric look suspiciously high, and there is no analysis of this. This seems to imply that the task has been technically solved, which is not the case. If the benchmarks have been saturated, the authors should either evaluate on a harder benchmark, or if not available select more advanced cases of the task, and see if the performance still holds (W3). Potentially also adding some case-studies where you analyze how the system performs, what it gets correct and what it gets wrong, and how it compares against other methods (W2). Although the idea is intellectually appealing, the authors do not show any examples of how the feedback looks like, and if there are any specific insights that the system has leveraged towards completing the task (W2). It is a bit sad that a paper is written entirely about molecular generation and optimization, and yet no single depiction of a molecule is shown. I think it's important to show examples, to demonstrate what the solutions given by your system really look like, if they are any good (W2). Simply showing a bunch of results on a table is not enough and at minimum lacks transparency (W2).

---

> > > ### Author Response · Authors · 2025-11-25
> > > **Follow-up on Our Reponses - Reviewer RJqH**
> > >
> > > Dear Reviewer RJqH,
> > > - **It has been almost a week since we posted our rebuttal responses, so we would like to follow up to see if you have any remaining concerns.**
> > > - Specifically, we have taken the following steps to address your comments:
> > >     1. We have released our code via an anonymous GitHub repository.
> > >     2. We have included additional ablation studies, case studies, examples of execution and feedback, as well as visualizations of the molecules before and after optimization.
> > >     3. We have provided further explanation regarding our choice of benchmarking, which follows prior work, and we have added a corresponding discussion in the paper.
> > >
> > > **Please let us know if you have any remaining concerns; otherwise, we kindly ask if you can reconsider your rating, as code release is not required by ICLR (we have also provided in rebuttal now) and our evaluation follows the standard benchmark established in prior works.**

---

### Note · Authors · 2026-01-03

I have read and agree with the venue's withdrawal policy on behalf of myself and my co-authors.